# Evolution of triclosan resistance modulates bacterial permissiveness to multidrug resistance plasmids and phages

Qiu E. Yang [1,5], Xiaodan Ma[1,5], Minchun Li[1,5], Mengshi Zhao[2], Lingshuang Zeng[1], Minzhen He[1], Hui Deng[2], Hanpeng Liao [1], Christopher Rensing[1], Ville-Petri Friman[3], Shungui Zhou [1]✉ & Timothy R. Walsh[4]✉

The horizontal transfer of plasmids has been recognized as one of the key drivers for the worldwide spread of antimicrobial resistance (AMR) across bacterial pathogens. However, knowledge remain limited about the contribution made by environmental stress on the evolution of bacterial AMR by modulating horizontal acquisition of AMR plasmids and other mobile genetic elements. Here we combined experimental evolution, whole genome sequencing, reverse genetic engineering, and transcriptomics to examine if the evolution of chromosomal AMR to triclosan (TCS) disinfectant has correlated effects on modulating bacterial pathogen (*Klebsiella pneumoniae*) permissiveness to AMR plasmids and phage susceptibility. Herein, we show that TCS exposure increases the evolvability of *K. pneumoniae* to evolve TCS-resistant mutants (TRMs) by acquiring mutations and altered expression of several genes previously associated with TCS and antibiotic resistance. Notably, *nsrR* deletion increases conjugation permissiveness of *K. pneumoniae* to four AMR plasmids, and enhances susceptibility to various *Klebsiella*-specific phages through the downregulation of several bacterial defense systems and changes in membrane potential with altered reactive oxygen species response. Our findings suggest that unrestricted use of TCS disinfectant imposes a dual impact on bacterial antibiotic resistance by augmenting both chromosomally and horizontally acquired AMR mechanisms.

Antimicrobial resistance (AMR) has emerged as a leading cause of morbidity and mortality, the report by Murray et al. estimated that approximately 4.95 million deaths could be associated with AMR across 204 countries and territories in 2019[1]. It is generally accepted that the global AMR burden is exacerbated by the overuse of antibiotics, resulting in the worldwide spread of antibiotic resistance genes (ARGs) across all "one-health" sectors[2,3]. Increasing evidence

suggests that exposure to environmental chemicals such as disinfectants[4,5], can also indirectly contribute to antibiotic resistance. For example, triclosan (TCS) is a bactericidal compound with broad-spectrum antibacterial activity, which is used in a variety of consumer products[6,7] and present in various human tissues and fluids[8]. Since the mode of action of TCS involves the blocking of lipid biosynthesis by binding to enoyl-acyl carrier protein reductase (FabI)[9,10], mutations in

[1]Fujian Provincial Key Laboratory of Soil Environmental Health and Regulation, College of Resources and Environment, Fujian Agriculture and Forestry University, Fuzhou 350002, China. [2]Fujian Key Laboratory of Traditional Chinese Veterinary Medicine and Animal Health, College of Animal Sciences, Fujian Agriculture and Forestry University, Fuzhou 350002, China. [3]Department of Microbiology, University of Helsinki, 00014 Helsinki, Finland. [4]Ineos Oxford Institute for Antimicrobial Research, Department of Biology, University of Oxford, Oxford OX1 3RE, UK. [5]These authors contributed equally: Qiu E. Yang, Xiaodan Ma, Minchun Li. ✉e-mail: sgzhou@fafu.edu.cn; timothy.walsh@biology.ox.ac.uk

this gene lead to bacterial TCS resistance. Increased TCS resistance has been linked to an elevation in bacterial resistance to other antibiotics in clinical settings[11–13]; however, the potential mechanisms involved in TCS resistance and its impact on mobile genetic elements (MGEs) remain unclear.

Bacterial AMR can evolve through de novo chromosomal mutations[14] or via horizontal acquisition of ARGs located in MGEs, such as plasmids[15,16]. Conjugative multidrug resistance (MDR) plasmids are of particular clinical concern as plasmids can transfer multiple ARGs at once, allowing taxonomically distinct bacteria to share similar, if not identical, ARGs[17,18]. In general, bacteria deploy several defense systems that can recognize foreign DNA delivered by plasmids and phages[19] and thereby inhibit the horizontal acquisition of MDR plasmids. However, this is contrary to the fact that MDR plasmids among pathogenic bacterial clones are highly prevalent[20]. Many recent studies have characterized the coevolution of bacterial host and plasmids under the stress of antibiotics[14,21–23], expanding our understanding as to how MDR plasmids evolve and persist in bacterial pathogens. Specifically, the fitness burden conferred by MDR plasmids can be ameliorated through downregulation of plasmid gene expression[24,25] or due to compensatory mutations in either host plasmid(s) or their chromosome[26–28]. Despite the importance of these studies, our knowledge remains limited as to whether bacterial genomic modifications induced by environmental chemicals would potentially affect bacterial permissiveness to MDR plasmids. In particular, while previous research has reported that TCS exposure resulted in bacterial resistance to TCS and antibiotic resistance[11,29,30], potential correlated effects of TCS resistance on the acquisition of AMR plasmids have remained relatively unexplored.

In this study, we explore bacterial adaptations under the exposure of TCS and how it modulates bacterial AMR via chromosomal mutations or altering conjugation permissiveness to MDR plasmids. To achieve this, we experimentally challenged a clinical strain *Klebsiella pneumoniae* Kp85 with increasing concentrations of TCS that ranged from sub-MIC (0.03 mg/L) to lethal concentrations (32 mg/L). Following 11-day TCS exposure, genomic sequencing, transcriptomics, and microbiological assays were applied to examine changes in TCS and antibiotic resistance, and potential genetic mechanisms that increased bacterial permissiveness to MDR plasmids as well as susceptibility to phages.

## Results

### Accelerated evolution of high-level TCS resistance in *K. pneumoniae* Kp85

To investigate the acquisition of TCS resistance, we performed an "evolutionary ramp" selection experiment[27], where the concentrations of TCS were doubled daily from a sub-inhibitory (0.03 mg/L, 1/16x MIC) to a lethal dose (32 mg/L, 64x MIC). A TCS-sensitive (MIC 0.5 mg/L) clinical *K. pneumoniae* Kp85 strain (Kp85anc)[31] was serially cultured on a 96-well plate for 11 days with increasing concentrations of TCS until all replicate populations ($n = 15$) failed to grow. All replicate populations were able to survive until day 6 (1 mg/L, 2× MIC), after which the number of surviving replicates decreased steadily (Fig. 1a, b and supplementary Table S3). To determine the resistance levels of TCS, a total of 13 TCS-resistant mutants (TRMs) were isolated from surviving replicates from day 7 to day 10, followed by the agar dilution. We found that exposure to TCS led to a 4- to 128-fold increase in TCS MICs (Fig. 1e and supplementary Table S4). Increased TCS resistance was also observed using confocal laser scanning microscopy where TRMs had higher proportion of living cells compared to the parental strain at 8 mg/L TCS (means of dead cells: 64.6% for Kp85anc vs 25.6% in d10-2 Fig. 1c, d). These results suggest that *K. pneumoniae* rapidly evolves resistance to TCS as shown previously[32]. As previous studies have correlated TCS resistance with antibiotic resistance[33–35], we also compared changes in antibiotic susceptibility of parental and TRMs using agar and broth microdilution. The

development of TCS resistance clearly increased the MIC of ciprofloxacin from 0.5 to 8 mg/L, as well as resulting in a four- and 16-fold increase in MICs for cefotaxime and fosfomycin, respectively (Fig. 1e and supplementary Table S4). While no change was observed to the other antibiotics tested (Supplementary Table S4).

### Genetic mechanisms of TCS resistance

To identify single nucleotide polymorphisms (SNPs) that could potentially identify the mechanism of TCS resistance, we chose five evolved clones from independent replicates at day 7 and two final evolved clones from day10 for further analysis (Fig. 2a). Notably, the majority of evolved clones carry mutations in *nsrR-like* gene and *ndh* (Fig. 2a and Supplementary Table S5), and both genes have been previously linked to bacterial stress responses[36]. *nsrR* belongs to the Rrf2 family of transcriptional regulators and responds to nitric oxide exposure, while the NADH oxidase-encoding gene *ndh* has been proposed to increase AMR by diminishing NADH oxidation and consequently increasing NADH concentration within cells[37]. Mutations in other genes included NADH-quinone oxidoreductase (*ndhC*) and RNA pyrophosphohydrolase (*rppH*) and TCS resistance mutations (*fabI*). Unexpectedly, either Δ*nsrR* or Δ*ndh* single-gene knockouts did not confer TCS resistance (Supplementary Fig. S1), suggesting that these single mutations were not fully responsible for increased TCS resistance.

We further compared the genome-wide transcriptional profiles of parental and one representative evolved clone, d10-2 with mutations in both *nsrR* and *ndh* genes, in the presence and absence of TCS (2 mg/L). Gene expression profiles of parental and evolved clone were modulated by TCS exposure; however, the relative number of differentially expressed genes was on average 2.4-fold (245 vs 130) higher in the parental strain compared to the evolved mutant (supplementary Fig. S2a, b), suggesting the evolved clone (d10-2) exhibits higher tolerance against triclosan exposure. Compared to the parental clone, the evolution of TCS resistance was associated with significant gene expression changes (Supplementary Fig. S2c, d), which were mainly associated with catalytic activity, cellular anatomic entity, metabolic processes, cellular structure and localization (Supplementary Fig. S3a). Specifically, the significant expression changes in several genes were directly associated with high-level resistance to TCS in the evolved clone d10-2 (Supplementary Fig. S4) and includes the overexpression of primary target of TCS, *fabI*, and *fabB*, and the downregulation of two stress regulators, *soxS* and *marA/R*, which have previously been shown to be important for survival at high TCS concentrations[30,38]. In addition, resistance to antibiotics observed in Fig. 1e were associated with transcriptional changes in *oqxB19-like*, *mgrB*, and *fosA5*, which are responsible for ciprofloxacin, colistin, and fosfomycin resistance, respectively. The evolution of TCS resistance has therefore resulted in cross-resistance to clinical antibiotics through differential gene expression affecting several independent bacterial functions.

### Downregulation of defense systems leads to increased conjugation permissiveness to AMR plasmids

In addition to the changes in the expression of genes involved in ABR, we also found that several bacterial defense systems were down-regulated in the evolved d10-2 mutant clone (Fig. 2b). These defense systems include type I-E CRISPR arrays (*cas123* and *casABCDE*), toxin-antitoxin systems (*ccdA/ccdB*, *abiEii* and *kacA/kacT*), type VI secretion system (T6SS, known to mediate bacterial competition, which may affect conjugation frequencies), *Klebsiella* capsular loci Wzi (have been reported to hinder DNA transfer and possibly constitute a barrier to plasmid transfer) and restriction-modification (RM) systems (HsdR, which are known to block invasive MGEs by identifying and eliminating foreign DNA i.e., plasmid and viral DNA)[19]. Specifically, three *cas* genes encoding type I-E CRISPR-associated endoribonucleases (*cas1e*, *cas2* and *casC*) were significantly downregulated in the evolved d10-2

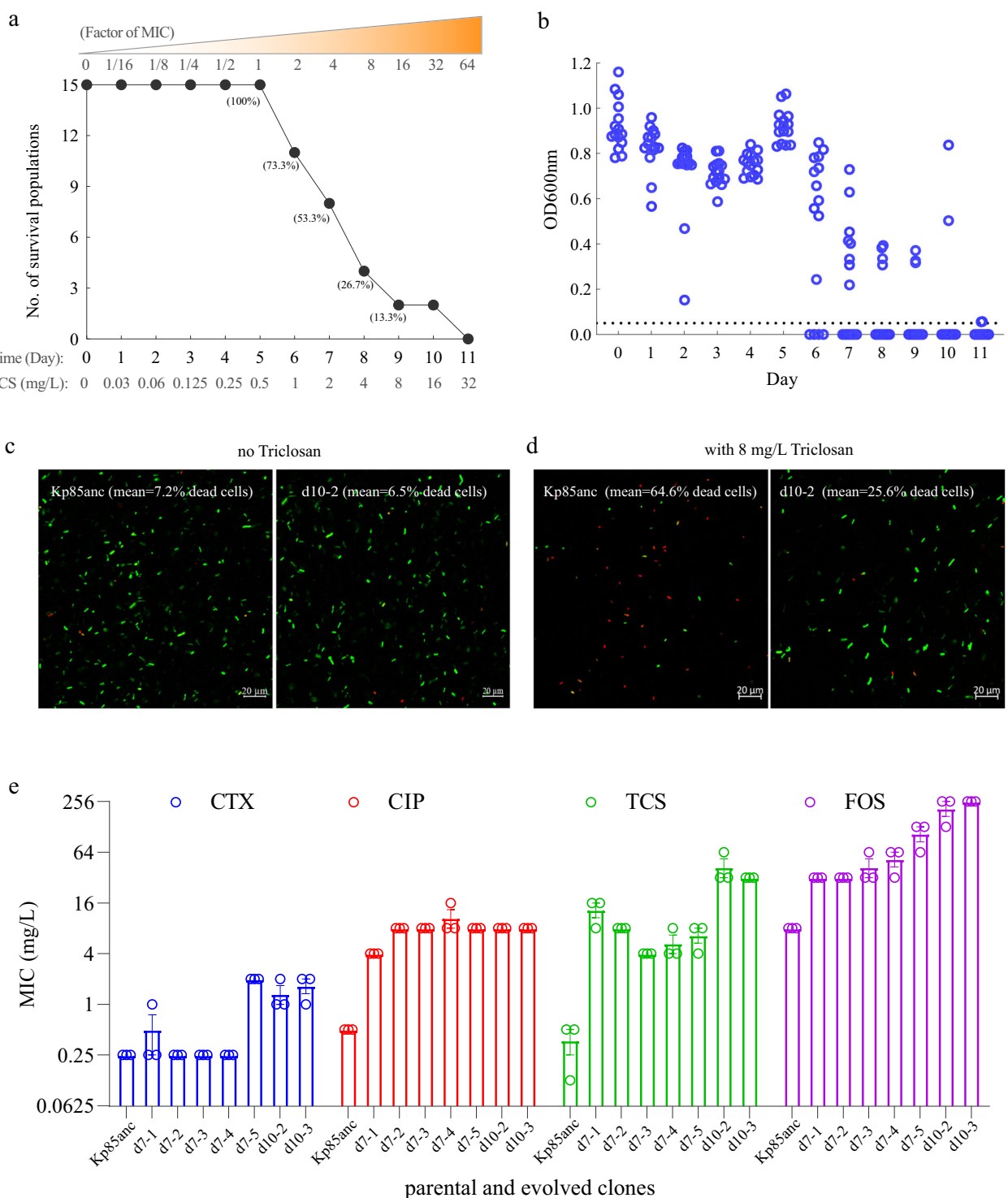

**Fig. 1 | Evolution of triclosan resistance in *Klebsiella pneumoniae* Kp85 strain.** **a** The number (and percentage) of surviving replicate populations ($n = 15$) at increasing triclosan concentrations from a low (1/16x MIC, 0.03 mg/L) to the highest used dose (64x MIC, 32 mg/L). **b** Daily population density profiles, representing the daily optical densities (final $OD_{600nm}$) of 15 clonal populations of each replicate. Bacterial $OD_{600nm}$ were measured before the daily transfer by a microplate reader (SpectraMax iD3, Molecular Devices). **c**, **d** Confocal laser scanning microscopy images of a representative evolved d10-2 and parental Kp85 clones in the absence or presence of 8 mg/L triclosan and stained with LIVE/DEAD staining ($n = 2$ biological independent samples). Alive and dead cells are shown in green and red colors, respectively, with scale bar of 20 μm. **e** The susceptibility of parental and TRMs against triclosan (TCS), cefotaxime (CTX), fosfomycin (FOS) and ciprofloxacin (CIP), determined by agar microdilution (each circle represents one biological independent repeat, $n = 3$). Data are presented as mean values ± SEM.

mutant clone ($\log_2$ FC < −1, $p < 0.001$), together with a slight decrease in the expression of other *cas* genes (*cas3*, *casA* and *casE*; $\log_2$FC > −1, adjusted $p < 0.001$, Fig. 2b). Furthermore, we observed the downregulation of *abiEii* toxin ($\log_2$ FC = −1.055, adjusted $p < 0.001$), which

is a bacterial abortive infection system that prevents phage infection and replication[39].

Since the downregulation of bacterial defense systems mentioned above could be linked to the persistence of MGEs[39,40], we

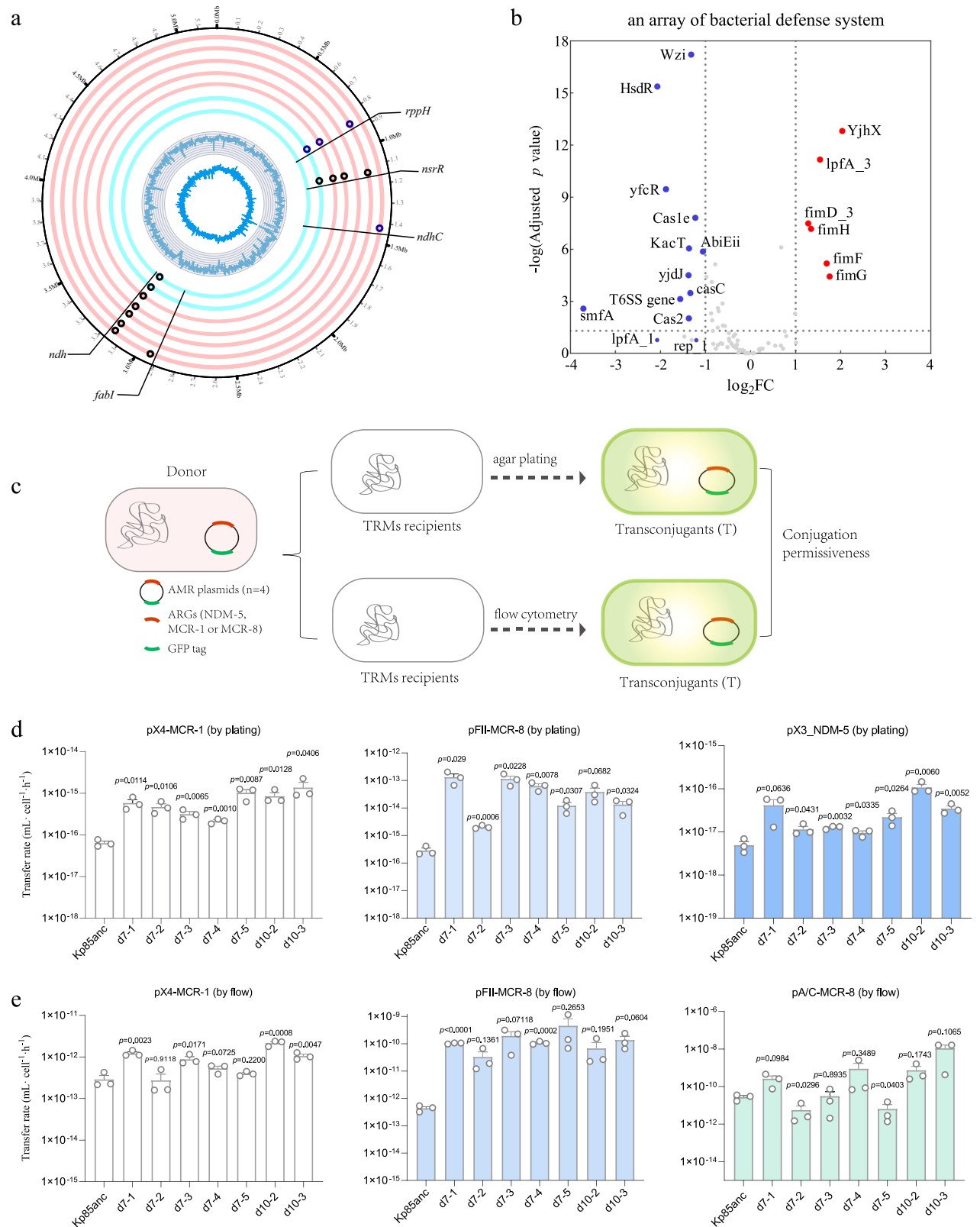

hypothesized that the evolved d10-2 clone is likely to possess an increase susceptibility to phages and permissiveness AMR plasmids. To test this hypothesis, we first investigated the ability of seven TRMs to acquire AMR plasmids by conjugation (termed "conjugation permissiveness"). For donors, we used four *E. coli* MG1655 strains carrying each of four different AMR plasmids including an IncX3 plasmid with a $bla_{NDM-5}$ gene (pX3_NDM-5)[41], an IncX4 plasmid with a

*mcr-1* gene (pX4_MCR-1)[14], an IncA/C plasmid with a *mcr-8* gene (pA/C_MCR-8)[42] and an IncFII plasmid with a *mcr-8* gene (pFII_MCR-8)[42]. To discount the possibility of differences in growth dynamics of donor, recipients and transconjugants, which could produce bias in measuring conjugation frequencies, the Simonsen's end-point method was applied[43]. As expected, significant variability in plasmid transfer was observed in different recipient-donor

**Fig. 2 | The genetic mutations and downregulation of bacterial defense systems leading to enhanced conjugation rates of AMR plasmids. a** Circular chromosomal view (CCV) of *K. pneumoniae* genome representing parental (gray ring) and seven evolved *K. pneumoniae* Kp85 evolved clones isolated at day 7 (pink rings) and day 10 (blue rings). From the outer to inner rings of the CCV, each ring displays the genomic variations in five evolved clones from five independent replicates at day 7 (pink rings: d7-1, d7-2, d7-3, d7-4 and d7-5), and two evolved clones isolated from day 10 (d10-3 and d10-2), the naming code is based on time points and numbers. For example, d10-1 name indicates that the evolved clone was isolated from day 10 and replicate 1. Red and blue dots overlaying rings represent nonsynonymous SNPs and indels (small insertions/deletions), respectively. The most inner rings represent GC skew and GC%, respectively. **b** The relative transcriptional changes in the gene expression of an evolved clone (d10-2) associated with bacterial defense systems. The edgeR method was applied for differential expression analysis of RNA-seq data. **c** The schematic representation of the conjugation experiments, in order to

compare the acquisition ability between parental strain Kp85 and TRM strains. Each donor strain contains one AMR plasmid, namely, *mcr-1* positive IncX4 plasmid (pX4_MCR-1), *mcr-8* positive IncFII plasmid (pFII_MCR-8), *mcr-8* positive IncA/C plasmid (pA/C_MCR-8) and $bla_{NDM-5}$ positive IncX3 plasmid (pX3_NDM-5). The transfer rates of four AMR plasmids were calculated by the Simonsen's end-point method (SM), while bacterial densities were measured by selective agar plating (**d**) and flow cytometer (**e**), respectively, indicating TRM strains had relatively higher plasmid transfer rates than the parental Kp85anc strain. All data were obtained in three independent experiments ($n = 3$, each circle represents one biological independent repeat) and data are presented as mean values ± SEM. The statistical analysis was performed by two-tailed *t*-test compared mean differences between each TRM strain to the parental strain (Kp85$_{anc}$) and the exact *P*-values were shown in each bar. The conjugation rates calculated by other methods were available in Supplementary Figs. S5 and S6.

combinations, with the majority of TRM strains showing higher conjugation rates compared to the parental strain for four of the AMR plasmids (Fig. 2d, one-way ANOVA, $P < 0.05$, Supplementary Table S6). The evolved clone d10-2 showed a significantly higher transfer rate than its parental Kp85anc strain for plasmids pX4_MCR-1 (two-tailed *t*-test, $P = 0.0114$) and pX3_NDM-5(two-tailed *t*-test, $P = 0.006$) in particular. As the donor strain and AMR plasmids were tagged with *mCherry* and *gfp* fluorescent genes, respectively, bacterial density can be measured by flow cytometer (Fig. 2c and Supplementary Table S1). Similar to the observations in Fig. 2d, this analysis revealed a wide diversity of conjugation frequencies across TRMs (one-way ANOVA, $P < 0.05$ except for the plasmid pFII_MCR-8, Supplementary Table S6). Moreover, the relative higher conjugation rates were observed in all TRMs for plasmid pFII_MCR-8, while four TRMs showed higher conjugation rates for the plasmid pX4_MCR-1 (Fig. 2e). Despite the variations using different AMR plasmids and calculations presented in Supplementary Fig. S5 and S6, our results repeatedly show that the majority of TRMs have enhanced AMR-plasmid conjugation rates (Supplementary Figs. S5 and S6 and Source data are provided as a source data file 1–4). Note that the transfer frequency of plasmid pX3_NDM-5 is below the detection limit of flow cytometer ($<10^{-5}$); subsequently, the method of antibiotic-selective medium was deployed. Plasmid pA/C_MCR-8 could not be distinguished from the recipient Kp85 by antibiotic selection; therefore, conjugation events for this plasmid were detected by flow cytometer only. In summary, the significantly increased conjugation rates were repeatedly observed in the majority of TRMs suggesting that the evolution of TCS resistance promotes bacterial conjugation permissiveness towards AMR plasmids.

## The *nsrR gene* modulates the transfer efficiency of AMR plasmids

Our *nsrR* knockout data suggest that this gene is not singularly responsible for the observed TCS resistance. However, *nsrR* encodes a transcriptional regulator belonging to the Rrf2 family that regulates DNA binding through effectors such as cellular redox status[44], which might modulate bacterial permissiveness to MGEs. To test this hypothesis, we first performed time-course conjugation experiments to examine the plasmid transfer into four *nsrR* variants; namely, Kp85anc, evolved d10-2 clone, Δ*nsrR*, and the Δ*nsrR* complementary clone, Δ*nsrR-c*. The deletion of *nsrR* gene had no detrimental effect on the growth rate compared to the parental strain (Kp85anc) in the absence of TCS (Supplementary Fig. S1). As shown in Fig. 3a, the transfer rates of pFII_MCR-8 were significantly affected by mating time and *nsrR* variant strains (two-way ANOVA, Time x strain, F (15,40) = 4.121, $P = 0.0002$). Compared to the parental Kp85anc, the evolved d10-2 exhibited higher conjugation rates of plasmid pFII_MCR-8 at four time points (Fig. 3a, two-tailed *t*-test: $P < 0.05$ at 4, 6, 8, and 16 h), which was also supported by other methods for calculating

plasmid conjugation rates (Supplementary Figs. S7 and S9). The Δ*nsrR* mutant displayed similar increases in plasmid acquisition (Fig. 3a, two-tailed *t*-test: $P < 0.05$ at 6, 8,16, and 24 h). Furthermore, we also tested the conjugation rates of three different AMR plasmids (pX3_NDM-5, pA/C_MCR-8, and pX4_MCR-1), showing higher transfer rates in both evolved d10-2 and Δ*nsrR* mutant, compared to the Kp85anc strain (Fig. 3b-c, two-tailed *t*-test, $P < 0.05$). Additionally, the conjugation rates measured by different methods (Supplementary Figs. S7–S8 and Source data are provided as a source data file 3–4)[43,45], consistently indicate that *nsrR* is, at least in part, responsible for increasing bacterial permissiveness to AMR plasmids.

## The *nsrR gene* modulates the susceptibility of *Klebsiella*-specific phages

To explore the effect of Δ*nsrR* on phage susceptibility, we conducted a large-scale phage infection model by challenging the above four Kp85 variants to a panel of 100 *K. pneumoniae*-specific lytic phages. The phage spot assay showed that a higher number of phages were observed in evolved d10-2 ($n = 54$) and Δ*nsrR* ($n = 50$) mutants, compared to Kp85anc ($n = 36$) (Supplementary Table S7 and Fig. S10). To further quantify differences in phage susceptibility, we determined how many phage particles could be produced in different *K. pneumoniae* host genotypes, using a subset of four phages. We found that the evolved d10-2 clone and Δ*nsrR* showed 1- to 6-$\log_{10}$ increases in plaque formation, indicating that reduced expression of defense systems facilitates efficient phage invasion and replication (Fig. 4a, b). Given the above, we further compared the kinetics of infection of fourteen different phages against the four Kp85 variants (Supplementary Fig. S11). We found that the growth of Kp85anc was the least affected by most of the phages, with better growth rates (mean growth reduction index of −22.02%). In contrast, the evolved d10-2 mutant clone was clearly inhibited by several phages (mean growth reduction index of 23.79%, $p = 0.0001$). Furthermore, the other seven TRMs generally showed an increased susceptible, depending on the identity of the phages ($p < 0.05$, Supplementary Fig. S12). These findings suggest that the evolution of triclosan resistance renders strain Kp85 more susceptible to phages.

Since plasmid conjugation is strongly affected by the recipient cells' membrane permeability and the reactive oxygen species (ROS) response[4,46], we compared the physiological effects (membrane potential) of Δ*nsrR* relative to the parent and TRM strains using specific fluorescent dyes. Both the Δ*nsrR* mutant and the evolved mutant clones displayed a constantly higher ROS response and membrane potential relative to the parental clone, regardless of the level of TCS exposure (adjusted $p = 3.8 \times 10^{-2} – 2.0 \times 10^{-4}$, using two-tailed *t*-test comparison, Supplementary Fig. S13). This could be further explained by the transcriptional changes in known ROS coping systems. Several ROS regulation genes were upregulated in the evolved d10-2 clone, relative to the parental clone (Supplementary Fig. S13a), including two

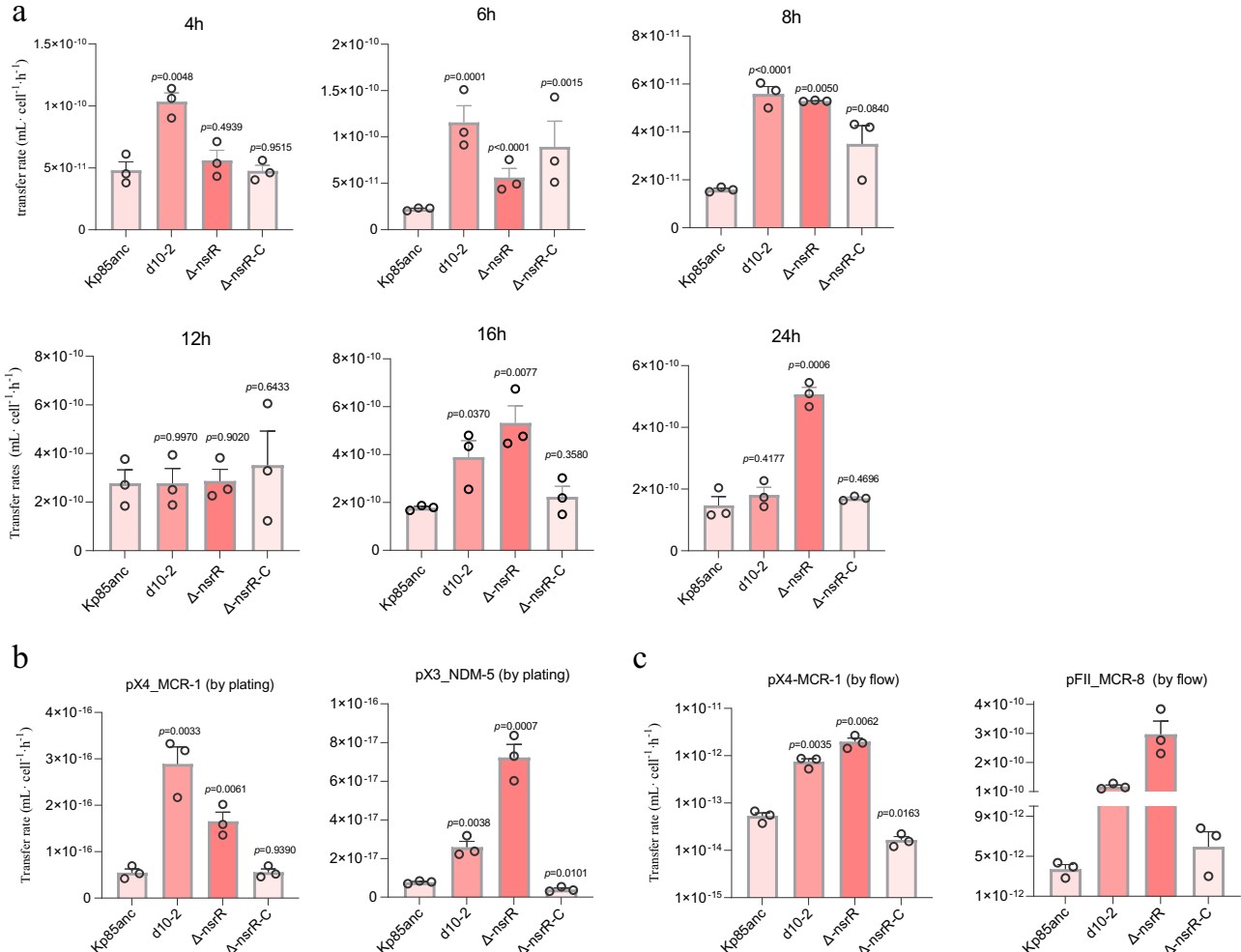

**Fig. 3 | Increased conjugation rates of AMR plasmids were observed in evolved d10-2 and ΔnsrR knockout clones across 24 h. a** The time-course conjugation rates were calculated by Simonsen's end-point method (SM), and the number of *gfp*-expressing transconjugants and *mCherry*-positive donor strain were measured by flow cytometry. **b**, **c** Conjugation rates calculated by Simonsen's end-point method (SM), and bacterial densities were determined by flow cytometry and selective agar plating, respectively. The statistical analysis was based on two-tailed *t*-tests comparing mean differences between each *nsrR* variatnt and the parental strain (Kp85anc). Three biological repeats were performed for each donor-recipient conjugation experiments. Conjugation rates calculated by other methods were available in Supplementary Figs. S7 and S8. The statistical analysis was based on two-tailed *t*-tests comparing mean differences between each mating group and the parental group (Kp85anc) (each circle represents one biological independent repeat, *n* = 3). Data are presented as mean values ± SEM and the exact *P*-values were shown in each bar.

thioredoxin reductase (*trxB* and *trxC*), a NADH oxidoreductase (*hcr*), a catalase family protein and an *oxyR*-regulated Alkyl hydroperoxide reductase C (*ahpC_2*). Thus, these findings suggest that *nsrR* plays an important role in modulating *K. pneumoniae* permissiveness to MDR plasmids by affecting their cell membrane potential.

## Discussion

Bacterial antibiotic resistance is commonly attributed to mutations in chromosome or acquisition of ARGs through horizontal transfer via MGEs. Herein we studied how prolonged exposure to non-antibiotic antibacterial disinfectant, TCS[37,47,48], shapes the evolution of *K. pneumoniae* resistance and subsequent permissiveness to MDR plasmids and phage infections. We show that TCS exposure selects for TRMs that are also resistant to common clinical antibiotics. We observed parallel mutations in *nsrR* and *ndh* genes, which were accompanied with consistent genomic changes and altered gene expression in the TCS *K. pneumoniae* clone relative to the parental strain. Interestingly, these changes were associated with altered expression of TCS and antibiotic target genes, suggesting transcriptional rewiring through unknown genetic or epigenetic mechanisms. Importantly, evolution of

TCS resistance increased *K. pneumoniae* permissiveness to several MDR plasmids and made the resistant mutants more susceptible to phage infections, via reduced expression of several bacterial defense systems also mediating changes in membrane potential and ROS response (the proposed mechanisms are illustrated in Fig. 5). These findings demonstrate that TCS-induced bacterial adaptation can promote horizontal transfer of MDR plasmids exacerbating the problem of global AMR. However, increased TCS resistance led to a trade-off in phage resistance, which could open new avenues for developing phage therapy to treat antibiotic-resistant infections.

Our genomic analysis showed that TCS resistance was associated with parallel mutations in both *nsrR* and *ndh* genes (Fig. 2a and Supplementary Tables S4 and S5), while mutations in previously characterized TCS target gene *FabI* (A21T)[9] were also observed in two evolved clones. In addition, two intergenic and insertion mutations were detected in the TRM clone d10-2, which were associated with hypothetical protein and an RNA pyrophosphohydrolase RppH gene, respectively (Supplementary Table S5). RppH mutants have been previously associated with different phenotypic properties, such as sensitivity to chemical stresses and increased membrane permeability[49], and

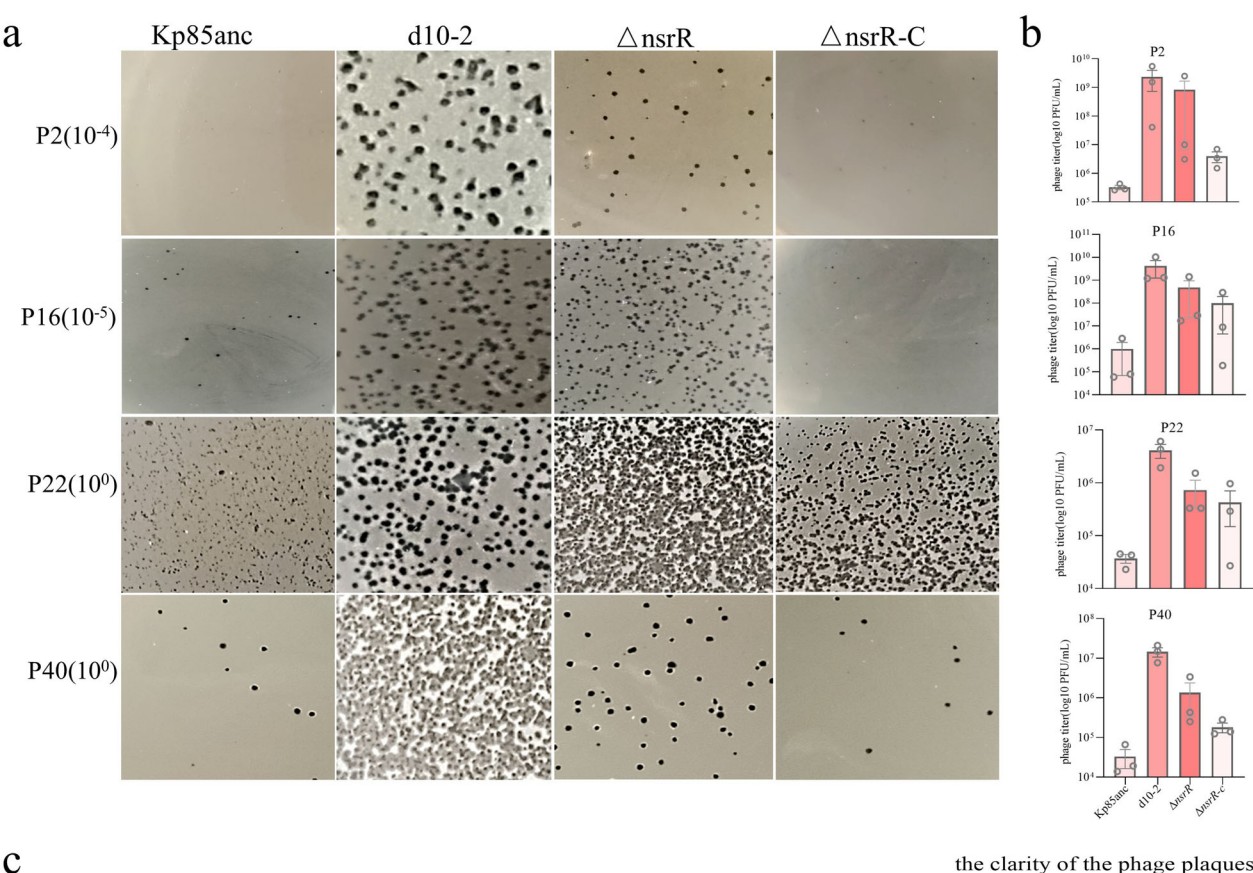

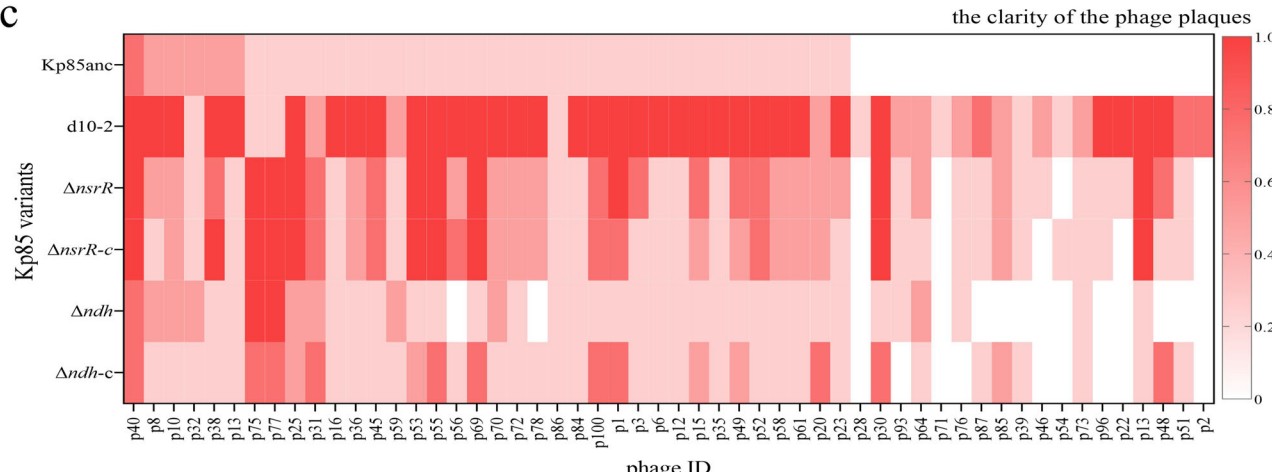

**Fig. 4 | Comparison of phage infectivity between parental, evolved and ∆*nsrR* knockout clones. a** Plaque morphologies of four selected phages (P2, P16, P22, and P40) against four Kp85 variants. The images were selected under the same dilution factor, and a higher numbers of phage plaques were consistently observed in evolved d10-2 and ∆nsrR strains, indicating the increased phage infectivity in both strains. **b** The efficiency of plaque formation of four selected phages (P2, P16, P22, and P40) against four Kp85 isogenic strains, determining by double-layer agar plate method. All data are based on three independent experiments (mean ± SEM, $n = 3$). The statistical analysis was performed using Mann–Whitney two-tailed $t$-test comparing mean differences between each group, while no statistical significance was shown ($p$ value > 0.05). Source data are provided as a source data file 5. **c** A heat map comparing the susceptibility of parental, evolved and genetically engineered knockout mutants to a panel of 54 lytic phages, where phage susceptibility was assessed by the plaque clarity based on spot-testing (ranging from 0= no visible infection to 1= fully clear plaques with no growth of resistant colonies, Supplementary Fig. S6). The ∆nsrR-c and ∆ndh-c indicate the complementary clones for knockout ∆nsrR and ∆ndh, respectively.

could have also affected the systemic changes in bacterial transcriptional profiles and antimicrobial susceptibility (Fig. 1e and Supplementary Fig. S2). We studied one representative TRM (d10-2) strain with both *nsrR* and *ndh* mutations in more detail, and found clear changes in gene expression of the previously determined TCS target gene, *fabI*[33], which was upregulated in the evolved TRM compared to the parental clone. Increased expression of *fabI* could have compensated the inhibition of fatty acid biosynthesis by TCS, and alleviate inhibition on

bacterial growth[50]. We also found that TRMs showed increased resistance to several clinically important antibiotics, including colistin, ciprofloxacin and fosfomycin. These effects were likely driven by downregulation (*mgrB*) or upregulation (*oqxB-like* and *fosA5* genes) of known antibiotic resistance genes. Cross-resistance to TCS and antibiotics (mainly chloramphenicol and tetracycline) have previously been reported in various bacterial pathogens including *Staphylococcus aureus*[11], *Pseudomonas aeruginosa*[51], *Salmonella* spp[52],

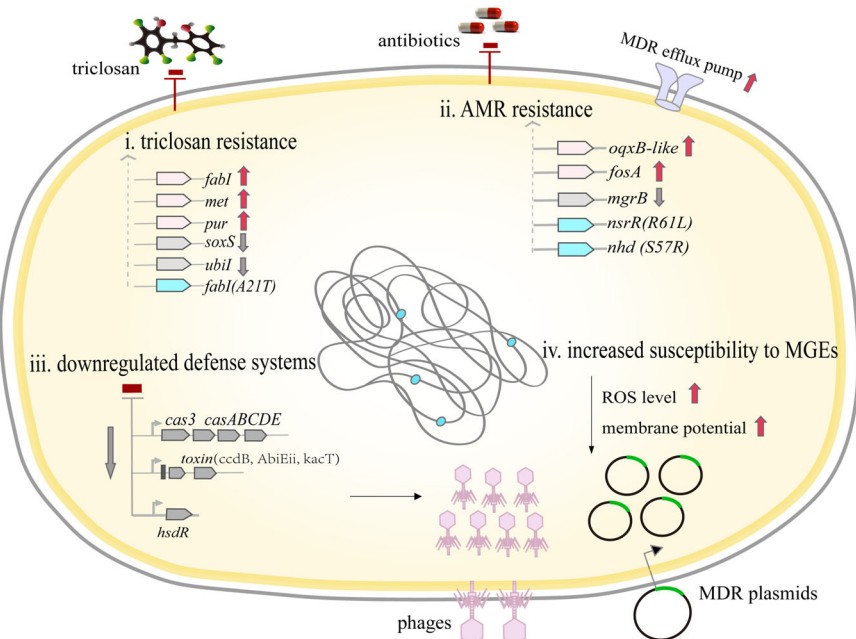

**Fig. 5 | The proposed mechanisms underlying increased antimicrobial resistance to triclosan and clinical antibiotics and increased *K. pneumoniae* permissiveness to multidrug-resistant plasmids and susceptibility to phage infections.** (i) The selection by triclosan leads to transcriptional changes in triclosan target genes and general stress response genes. (ii) At the genetic level, triclosan resistance is associated with mutations in *nsrR* and *ndh* genes, resulting in cross-resistance to clinical antibiotics. Cross-resistance is also partly due to expressional changes in antibiotic target genes, including upregulation of *oxqB*-like efflux pump and changes in fosfomycin and colistin target genes (*fosA* and *mgrB*, respectively). (iii) Evolution of triclosan resistance is coupled with downregulation of several bacterial defense systems, including type I CRISPR-Cas system, several toxin-antitoxin systems and R-M system, and changes in cell membrane potential. (iv) As a result, evolution of antimicrobial resistance turns *K. pneumoniae* more permissive to several multidrug-resistant plasmids and increases their susceptibility to several lytic phages.

and mechanistically, these positive trait correlations have been linked with AcrAB efflux pump activity and changes in cell membrane.

Importantly, we show that TCS-resistant *K. pneumoniae* mutants were more susceptible to conjugation by MDR plasmids (Fig. 2c-e and Fig. 3). Notably, increased plasmid transfer was observed in the absence of TCS, suggesting that increased conjugation rates were not driven by potential cross-resistance benefits provided by the MDR plasmids. Mechanistically, it is likely that the increased plasmid permissiveness could be linked to downregulation of several bacterial defense systems, including the type I-E CRISPR-cas system, toxin-antitoxin systems and the restriction-modification systems (Fig. 3a). Recent studies have proved that three bacterial defense systems, the Wadjet in *Bacillus subtilis*[47] and DdmABC/DdmDE in *Vibrio cholerae*[48], can specifically recognize foreign DNAs to protect its host against plasmid transfer, suggesting that the downregulated bacterial defense systems we observed could also be potentially linked to enhancing bacterial permissiveness towards plasmid DNA. Moreover, the downregulation of defense systems in TRMs enhanced their susceptibility to infections by lytic phages. This trade-off was clearly mediated by the *nsrR* gene as Δ*nsrR* showed increased susceptibility to a range of different phages. These findings are in line with previous studies wherein phages susceptibility represents an evolutionary trade-off in MDR bacteria[53,54]. The *nsrR* gene was also associated with increased bacterial membrane permeability, which could have made the evolved resistant cells more permissive to plasmids. Specifically, knocking out the *nsrR* gene increased bacterial ROS response and proton motive force (Supplementary Fig. S9), which are required for DNA exchange, thereby promoting the acquisition of MDR plasmids[4,46,55]. NsrR is a regulatory protein and plays an important role in performing several biological functions in many microorganisms, especially with pathogenic bacteria. Its most notable function is to serve as a global regulator of bacterial nitric oxide (NO) sensing repressor[36,56,57], regulating

the genes involved in both NO detoxification and NO damage repair. Other targets of regulation by NsrR in *Escherichia coli* include the promoter regions of *fliA*, *fliL*, and *mqsR*, which encode proteins involved in bacterial motility and biofilm development[58]. However, the limitation of this finding is that we could not generate the key point mutation in *nsrR* gene, which would precisely replicate the genotype observed in the TRMs clones. Whilst further studies are required to delineate specific molecular mechanisms, our findings suggest that NsrR also regulates antimicrobial resistance and defense systems against MGEs.

In conclusion, our findings demonstrate that the evolution of TCS resistance can contribute to the global AMR burden via cross-resistance to clinically important antibiotics and by enhancing permissiveness to MDR plasmids. Importantly, TCS resistance was associated with increased susceptibility to phage infections, which could open new avenues for using phage therapy to treat antibiotic/TCS-resistant *K. pneumoniae* infections. Increased phage susceptibility could also partly explain the efficacy of phage-antibiotic combination treatments recently used to treat a 30-year-old patient with a fracture-related pan-drug-resistant *K. pneumoniae* infection[59]. Further studies on how common bactericidal disinfectants shape the evolution of chromosomal and horizontally acquired antimicrobial resistance is of critical importance to develop alternative therapies to curb the worldwide spread of ARGs.

## Methods
### Bacterial strains, plasmids, phages and growth conditions
The bacterial strains, plasmids, and phages used in this study are listed in Supplementary Table S1 and Table S7. The *K. pneumoniae* Kp85 and *E. coli* MG1655 strains were grown in Luria-Bertani (LB) broth (Sigma-Aldrich) at 37 °C with shaking (150 rpm) or on LB agar plates. It is noted that strain Kp85 was isolated from discarded feces of a hospitalized

patient[31], and that consent was therefore not required. TCS and antibiotics were added at the following concentrations for both conjugation experiments and genetic tool editing in the experiments: 0.5 to 64 mg/L TCS, 30 mg/L kanamycin, 1 mg/L meropenem, 3 mg/L colistin, 50 mg/L apramycin, 50 mg/L spectinomycin, and 8 mg/L tigecycline. Bacterial densities were measured as the optical density at 600 nm ($OD_{600nm}$) every hour by SpectraMax iD3 (Molecular Devices, USA) when conducting growth curve measurements. The *K. pneumoniae*-specific phages used in this study were previously isolated from urban wastewater treatment plants.

## Experimental evolution setup

The experimental evolution experiment was carried out using previously established 'evolutionary ramp' method, where antibiotic concentration is increased in time with following modifications[14]. Briefly, wild-type *K. pneumoniae* Kp85 were first grown on LB agar medium at 37 °C for overnight. To ensure reproducibility, we used 15 independent replicate cultures and each population was started from a different colony. These replicate cultures (approximately $10^6$ cfu/mL) were propagated on 96-well microtiter plates where the inner 15 wells were used for bacterial cultures and the outer wells for media blank measurements and contamination controls. The plates were incubated for 22 h after homogenized replicate cultures were serially diluted 1:400-fold into fresh LB medium on 96-well plates with TCS. The concentration of TCS was doubled daily from the very low dose (1/16 MIC, 0.03 μg/ml) to a very high dose (64× MIC, 32 μg/mL). Before each transfer, the optical density was measured and the experiment was carried out until all replicates had gone extinct (*i.e.*, $OD_{600nm}$ was less than 0.05). This resulted in total of 11 serial transfers during the selection experiment. Replicate populations were streaked on Chromogenic UTI agar plates every day to ensure no cross-contamination occurred during the experiment (Kp85 appears as blue colonies on this selective medium). A single evolved colony from different replicates was selected and subjected to TCS resistance determination. In total, 13 evolved Kp85 clones were isolated (Supplementary Table S3) and preserved at −80 °C in 30% glycerol solution for further study.

## Determination of minimum inhibitory concentration (MIC) for TCS and antibiotics

Minimum inhibitory concentrations (MICs) of TCS and antibiotics (for one parental and TRMs) were determined by agar dilution method, according to Clinical and Laboratory Standard Institute (CLSI)[60]. The bacterial culture was diluted in sterile saline solution and the bacterial density adjusted to approximately $10^6$ CFU/mL. The multipoint inoculum tool was dipped in prepared bacterial solution and inoculated on the surface of Mueller–Hinton (MH) agar plate containing serial concentrations of antibiotics. The agar plates were dried for 5 min and the incubated at 37 °C for 16–20 h before bacterial growth was observed. The minimum concentration of drug in the agar plate that inhibited the bacterial growth was determined as MIC[61].

A standard broth microdilution method was used to determine the MIC for colistin[60]. The concentrations of colistin were 1.5-fold diluted in fresh MH broth on 96-well microtiter plates, resulting in final concentrations of colistin from 0 to 6.4 mg/L. The parental and evolved strains were grown in LB medium supplemented with appropriate antibiotics at 37 °C for overnight. Following overnight incubation, approximately $1 \times 10^6$ cells were inoculated into each well of the 96-well microtiter plate with colistin. Three independent replicates for each strain and the corresponding control were used. The top and bottom row in the 96-well plate were filled with MH broth to obtain the background $OD_{600nm}$ value of the growth medium. Plates were incubated at 37 °C without shaking for 20–24 h, until $OD_{600nm}$ values were measured in a microplate reader (SpectraMax iD3). After background subtraction, MIC was defined as the lowest concentration of colistin where the $OD_{600nm} < 0.05$.

## Whole genome sequencing and bioinformatic analysis

To identify the polymorphisms associated with TCS resistance evolution, we sequenced 13 TRMs obtained from day 7 (4× MIC) to day 10 (16× MIC), exhibiting high resistance to TCS. Bacterial DNA was extracted using genomic DNA (gDNA) extraction kit (TIANGEN, China) following the manufacturer's protocol. The concentration of each gDNA sample was determined using a Qubit Flex System (Invitrogen, United State). The purified gDNA samples were subjected to Illumina NovaSeq sequencing. The parental Kp85anc was also sequenced with the Oxford nanopore MinION platform to obtain high-quality reference genome. Raw Illumina reads were filtered with Trimmomatic v. 0.38.1[62] with default parameters. Raw Nanopore reads were trimmed using Mecat2 and assembled using SMRT link v5.1.0. Hybrid genomic assembly between Nanopore and Illumina sequencing reads was performed by using unicycler v. 0.4.8[63] (https://github.com/rrwick/Unicycler). The resulting hybrid-assembled reference genomes was annotated using Prokka v. 1.14.6 (Galaxy version)[64]. Genomic mutations were identified using breseq v. 0.34.0 in the polymorphism prediction mode[65]. All sequencing data are available in the NCBI sequence Read Archive, with accession numbers (BioProject accession no: PRJNA937637, and BioSample accession No. SAMN33774534 and SAMN33604226 with 14 SRA fastq data for the parental and 13 TRMs).

## RNA-Seq Analysis

To compare how TCS exposure changed bacterial gene expression, we conducted RNA-Seq analysis for the parental (Kp85anc) and one representative evolved *K. pneumoniae* strain (d10-2) in the absence and presence of TCS. Three replicates were conducted for each condition, and the data are represented as mean expression of each gene ($n = 3$). In brief, exponentially growing cultures ($OD_{600nm} = 0.5$) of the parental (Kp85anc) and evolved (d10-2) strains were treated with or without 2 mg/L TCS (4x MIC) for 2 h before bacterial pellets were collected by centrifugation ($5000 \times g$, 4 °C,10 min), immediately frozen down using liquid nitrogen and stored at −80 °C before being processed for RNA extraction. The total RNA was prepared using the RNA Extraction Kit (QIAGEN, Germany) and rRNA was removed using Ribo-Zero rRNA Removal Kit (Epicentre, United States), following the manufacturer's protocol. Sequencing libraries were generated using the NEBNext Ultra Directional RNA Library Prep Kit for Illumina (New England Biolabs, United States), followed by sequencing on an Illumina HiSeq 2500 platform. Bowtie2[66] was used to trim and clean the sequence data and the reads were mapped to each gene using the parental reference genome. Differential gene expression analysis was conducted by the edgeR R package (v3.40.2)[67], and the results were deposited into the NCBI database (BioProject accession no: PRJNA932187, and BioSample accession No. SRR23356413 to SRR23356423).

## Genomic deletion and complementation of *nsrR* and *ndh* genes

The deletion of *nsrR* and *ndh* genes was carried out by a CRISPR-based two-plasmid system, pCasKP and pSGKP, as described in a previous study[68]. In brief, the pCasKP-apr plasmid was transferred into parental *K. pneumoniae* clone (Kp85anc) by electroporation, yielding the pCasKP-apr positive Kp85anc strain. The 20-bp spacers (N20) were inserted to plasmid pSGKP by reverse PCR, resulting the N20-containing plasmid pSGKp-ndh-N20 and pSGKp-nsrR-N20, respectively. All primers and plasmids are listed in Supplementary Tables S1 and S2. Next, the 200 ng N20-containing plasmids pSGKp-ndh-N20 or pSGKp-nsrR-N20 were co-transformed with 300 μM ssDNA (donor repair template) into Kp85anc strain hosting the pCasKP-apr plasmid by electroporation. The cells were plated on LB agar plate added with 50 mg/L rifampin and 30 mg/L apramycin to isolate transformed knockout strains with given antibiotic resistance genes. The successful deletions were confirmed by PCR and sequencing.

With complementation, the full-length fragments of *nsrR* and *ndh* genes with its own promoters, were amplified from Kp85anc clone, and inserted to plasmid pHSG299 by Gibson assembly cloning kit (NEB, United States). The resulting plasmids pHSG299:*nsrR* or pHSG299:*ndh* were then transferred to Δ*nsrR* and Δ*ndh* knockout strains via electroporation. The successfully complemented strains were confirmed by PCR and sequencing.

## The determination of conjugation rates and frequencies

Since plasmid conjugation rates often depend on several key parameters, including the time of measurement, the initial population density or the initial ratio of donor-recipient populations[45], we applied the Simonsen's end-point method[43] and the extended Simonsen model (ASM) described in Huisman et al.[45], where the conjugation and growth assays were conducted in a 96-well microtiter plate. In brief, the overnight culture of each strain was diluted by 1:100 and re-grown into fresh LB medium for 2 h at 37 °C and 200 rpm, followed by dilution of each strain to approximately $10^6$ CFU/mL in fresh LB medium. The donor and recipient strains were then mixed 1:1 and the mating performed in LB broth at 37 °C. The $OD_{600nm}$ of monoculcure (D, R, T) and mixed conjugation cultures were measured every hour by SpectraMax iD3 (Molecular Devices, United States). The bacterial density was enumerated either by selective agar plating or flow cytometry, as described in the below method (i) and (ii), respectively. Finally, we calculated the conjugation rate (mL· cell$^{-1}$ h$^{-1}$) using the below equations:

$$\text{growth rate }(\psi) = \frac{\ln\left(\frac{OD_b}{OD_a}\right)}{T_b - T_a} \tag{1}$$

$$\text{transfer rate}(\gamma) = \psi \cdot \ln\left(1 + \frac{T \cdot N}{R \cdot D}\right) \cdot \left(\frac{1}{N - N0}\right) \tag{2}$$

In Eq. 1, the growth rate ψ is determined by the optical density of the mating culture at different time points, where $T_b$ and $T_a$ indicate the times (h) at which $OD_b$ and $OD_a$ were collected. In Eq. 2, the conjugation rate γ is estimated from four key determinations: population bacterial growth (ψ); the end-point cell density of D, R and T, standing for donors, recipients, and transconjugants respectively; initial population density $N_O$; and N = D + R + T is the total population density at the time of sampling (end-point). The conjugation rates also can be calculated via a Shiny web application (https://ibz-shiny.ethz.ch/jhuisman/conjugator/)[45]. All raw conjugation data was available in Source data files 1–4.

It is important to note that two different methods were employed to measure bacterial density, antibiotic agar plating and flow cytometry. (i) antibiotic agar plating: at the time of sampling (the initial and end-point), bacterial cultures were diluted and plated on selective UTI agar plates containing various antibiotic combinations, such as 1 mg/L of meropenem and 8 mg/L of tigecycline for transconjugants with plasmid pX3_NDM-5, and 3 mg/L of colistin and 8 mg/L of tigecycline for transconjugants with two colistin resistance plasmids (pX4_MCR-1 and pFII_MCR-8). While 8 mg/L of tigecycline was used for selecting recipient strains. The successful transconjugants were further verified by PCR targeting $bla_{NDM-5}$, *mcr-1* and *mcr-8* genes, respectively.

(ii) bacterial densities were measured by flow cytometry: at the time of sampling (the initial and end-point), bacterial cultures were diluted and analyzed on an Attune NxT flow cytometer (ThermoFisher, USA) with bacterial cell size and fluorescent signal threshold settings. Fluorescent controls of *E. coli* MG1655::mcherry, *E.coli* DH5α::gfp and no fluorescence recipient community were prepared to set PMT voltages as previous described[69] and appropriate gating the green

fluorescent transconjugant cells and red fluorescent donor cells were excited by laser at 488 nm and 561 nm, respectively. The Gating strategy was provided in supplementary Fig. S14. For each mating mixture, approximately 50 μL were collected and the transfer frequency was roughly calculated by the percentage of *gfp*-expressing transconjugants present in the mixed culture.

Therefore, the transfer frequency of AMR plasmids also can be determined using the two classic calculation formulas provided below: conjugation frequency per recipient and conjugation frequency per cell were calculated by:

$$\text{transfer frequency per recipient}(T/R) = \frac{CFU(T)}{CFU(R)} \tag{3}$$

$$\text{transfer frequency per cell}(T/N) = \frac{gfp \text{ expressing cell counts}(T)}{\text{total cell counts}(N)} \tag{4}$$

Where in Eq. (3), $CFU_{(T)}$ and $CFU_{(R)}$ are the transconjugants/recipients colony-forming units obtained from the number of colonies on the selective agar plates (corrected by the dilution factor), respectively. While in Eq. (4) the number of bacterial cells was enumerated by flow cytometry.

## The time-course conjugation experiments between *nsrR* variants and plasmid pFII_MCR-8

To investigate the conjugation permissiveness of four *nsrR* variants (Kp85anc, d10-2, Δ*nsrR* and Δ*nsrR-c*) towards a representative plasmid pFII_MCR-8, the conjugation frequency was measured at time-course manner and bacterial density was measured by flow cytometry, as described above (ii). In addition, the transfer dynamics were also visualized by confocal laser microscopy as described by the previous study[48]. In brief, donor and recipient strains were mixed 1:1 and mating was performed in LB broth 37 °C. The transfer of plasmid pFII_MCR-8 was examined at 2, 4, 6, 8, 12, 16 and 24 h. At each sampling point, 100 μL of bacterial suspension were immobilized by mixing with 1.5% (wt/vol) agarose in PBS, covered with a microscope cover glass. The transfer of AMR plasmids was then examined using confocal laser microscopy (Zeiss LSM880, Germany). Images were analyzed and prepared for publication using ImageJ (version 1.48 v, http://imagej.nih.gov/ij). The conjugation frequency per donor was calculated by the formula below:

$$\text{transfer frequency per donor}(T/D) = \frac{gfp \text{ expressing cell counts}(T)}{mCherry \text{ expressing cell counts}(D)} \tag{5}$$

## Determination of phage susceptibility and efficiency of plaque formation

Phage host range spectrum was determined using spot lysis assay as previously described[70]. In brief, each phage lysate (N = 100) was spotted (5 μL) in duplicates onto soft agar overlays of four Kp85 genotypes, respectively. The phage host range spectra were determined according to morphology and clearness of the formed plaques, which were classified into five groups: (i) complete clearing (defined value of 1.00); (ii) clearing throughout but with faintly hazy background (defined value of 0.75); (iii) substantial turbidity throughout the cleared zone (defined value of 0.5); (iv) a few individual plaques (defined value of 0.25); (v) no plaques (defined value of 0.00).

To further quantify the phage replication on different host genotypes, the efficiency of plaque formation (PFU/mL) was measured using double-layer plaque assay as previously described[70]. Kp85 variants were inoculated in LB medium and incubated overnight at 37 °C.

Double-agar-layer assays were carried out using 0.6% LB agar as top agar, and 1.5% LB agar at the bottom. Phages were serially diluted in LB broth ($10^0$ to $10^{-7}$) and 100 µL of each dilution was mixed with bacterial hosts (50 µL), and incubated 10 min at 28°C. Bacteria and phage were mixed well into 0.6% LB top agar (47 °C), followed by plating on the bottom agar. Plates were imaged after overnight incubation at 37°C (~20–24 h) and plaque forming units (PFU/mL) were calculated by dividing the number of plaques for each tested Kp85 strains by the dilution factor.

## Phage infection kinetics

To compare the dynamics of phage infection, overnight cultures of four Kp85 clones were diluted in fresh LB broth (1:200) and grown at 37°C for 2 h with vigorous shaking (200 rpm). Twenty µL of each culture was mixed with phage solution in equal volumes and then moved to the wells of 96-well plates containing 160 µL of fresh LB broth. The concentrated phage lysates were prepared by infecting their original isolation host strains at 37°C with shaking for 6 h. Phage lysates were then centrifuged and passed through 0.2 µm sterile filters for three times to separate them from phages. Phage titers were determined using double-agar-layer assays as described above, and approximately $10^8$ PFU/mL of each phage were used to infect different Kp85 clones. Bacterial growth was measured using SpectraMax iD3 plate reader (Molecular Devices) at 1 h intervals for 24 h at 37°C with shaking. The growth reduction index ($t = 24$ h) represents the mean percentage growth reduction by each phage treatment based on three biological replicates: growth reduction index = $[(OD_{600nm(no\ phage)} - OD_{600nm(with\ phage)})/OD_{600nm(no\ phage)}] \times 100$.

## Measurement of bacterial membrane potential and ROS response

Exponentially growing cultures of different Kp85 variants (Kp85anc and Kp85evo and *nsrR* and *ndh* knockout strains) were treated with TCS (0, 0.25, 1.0 mg/L) for 2 h, for 20 min before cells were centrifuged at 10000 rpm for 10 min. Cell pellets were washed twice with PBS buffer before they were resuspended in JC-1 working solution included in the Mitochondrial membrane potential assay kit (with JC-1, Beyotime Institute of Biotechnology, China), following the manufacturers' protocol. The fluorescence intensity was measured by a microplate reader (SpectraMax iD3) and the detection parameters for JC-1 monomer and aggregate fluorescence were excitation at 490 nm and 590 nm wavelengths, respectively. The ratio between the red (JC-1 aggregates) and green fluorescence (JC-1 monomers) cells was used to determine the bacterial membrane potential as described previously[71].

The accumulation of intracellular ROS was measured using an ROS assay kit (Beyotime Institute of Biotechnology, China). A culture lacking the fluorescent dye was included as a control for autofluorescence. After TCS treatment for 2 h, the fluorescence intensity of bacterial cells was measured using a SpectraMax iD3 microplate reader (Molecular Devices) and the detection parameters for fluorescence were excitation and emission at 488 nm and 525 nm wavelengths, respectively. All tubes with cultures were wrapped with aluminum foil to avoid light. Each data point represents the average of four independent measurements.

## Statistical analysis

Data analysis was performed using GraphPad Prism (8.3.1). All conjugation data shown in plots are represented as mean of three replicates ± SEM, and exact number of independent replicates for each experiment is stated in their respective figure legends. Significant differences were determined by one/two-way ANOVA or two-tailed *t*-tests, as appropriate. The significance level was set to 0.05 and the exact *P* values were shown in the figures. For conjugation frequency experiments, analyses were performed on log-transformed data.

## Reporting summary

Further information on research design is available in the Nature Portfolio Reporting Summary linked to this article.

## Data availability

All data generated or analyzed during this study are included in the main text and its supplementary files. The DNA sequencing data generated in this study have been deposited in the NCBI sequence Read Archive under accession numbers (BioProject accession no: PRJNA937637, and BioSample accession No. SAMN33604226 to SAMN33604226). The RNA-seq data generated in this study have been deposited in NCBI database with accession number (BioProject accession no: PRJNA932187, and BioSample accession No. SRR23356413 to SRR23356423). Source data are provided with the paper. Source data are provided with this paper.

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

## Acknowledgements

We are grateful to Prof. Wang Yang (China Agriculture University) and Dr. Zhai Wei-shuai for providing the parental strain Klebsiella pneumoniae Kp85anc. This work was supported by National Key R&D Program of China, grant number 2021YFD1900400 to Q.E.Y; the National Science Foundation of China, grant number 32100150 and 42277436 to Q.E.Y; and the Science Foundation of Fujian province, grant number 2021J01116 to Q.E.Y.

## Author contributions

Q.E.Y. was responsible for conceptualization, data analysis, methodology, writing the original draft and reviewing and revising it. X.D.M performed evolution experiment, genetic editing, and interpreted results. M.C.L and M.S.Z contributed to gene deletion and phage susceptibility. L.S.Z and M.Z.H contributed to plasmid constructs and conjugation experiments. H.D, H.P.L, and C.R reviewed the manuscript. V.P.F was involved in reviewing and editing the manuscript. S.G.Z and T.R.W were involved in conceptualization and supervision and reviewing and editing the manuscript. All authors commented on and approved the manuscript for submission.

## Competing interests

The authors declare no competing interests.
