## [Peer Review File · Nature Communications]

Editorial Note: This manuscript has been previously reviewed at another journal that is not operating a transparent peer review scheme. This document only contains reviewer comments and rebuttal letters for versions considered at *Nature Communications*. Mentions of the other journal have been redacted.

REVIEWER COMMENTS

Reviewer #1 (Remarks to the Author):

I had previously reviewed the manuscript by Yang and collaborators when it had been submitted to [Redacted]. This new version represents a clear improvement and the authors have performed additional experiments, redone analyses and deeply rewritten the manuscript. However, the general impression of lack of rigour is still present, although to a lesser extent than before. I list below the points that need to be addressed.

1. The novelty of the paper mainly resides in the increased conjugation rates and phage sensibility of the populations evolved in presence of triclosan and the exploration of a potential mechanistic explanation for this pleiotropic effect. The part on TCS resistance evolution and evolution of cross-resistance to other antibiotics could be shortened to get more rapidly to the most interesting and newest point. This finding should also be discussed in the context of other recent studies on similar questions (e.g. Xuan et al. 2022 doi: 10.1128/spectrum.01356-22; Chen et al. <https://www.nature.com/articles/s41598-017-06688-w>; Chan et al. <https://www.nature.com/articles/srep26717>).

2. Whole genome sequencing: there is still a lot of confusion about the number and the identity of the clones sequenced. The results section says “five clones from day 7” and then mentions “other clones from day 8 and 10. Figure 2 legend says “five from day 7 + two from day 10”. Methods says “13 TRMs from day 7 to 10” which corresponds to the content of table S4. Even more importantly, it is still not clear to me whether clones from different days are sampled from independent experimentally evolved populations. The interpretation of finding the same mutations in clones from different time points, although informative in both cases, is quite different depending on whether they come from the same population or not.

3. L225-226: where does this affirmation come from? Please refer to a figure, a table, a statistical test + “singularly responsible” should “responsible alone” + as mentioned in my first review, the mutations in *nsrR* in exp evol populations are non-synonymous SNPs so there was no reason to expect that KO would recapitulate the effect of these mutations.

4. L114-115 “re-exposed to 2 and 4mg/L of TCS in short-term growth assays.” + FigS1: I do not understand the interest of getting this growth curves as the clones have been isolated at days where the TCS concentration was equal or superior to 4mg/L, so it is clear that they are able to grow at these two concentrations. The only piece of info from figure S1 is that TCS resistance has a cost in absence of TCS (TRMs grow less than the ancestor at 0mg/L), but this is not mentioned neither in the results nor in the discussion. More specifically, on L114-115, no results are actually given for this specific assay and as the following sentence is about MIC, it seems that the MIC have been obtained by this “short-term growth assay” but this is technically not possible (broader concentration range needed), so it is inducing confusion.

Minor points

These points are minor but all together they contribute to the impression of lack of rigour.

1. The methods are not presented in the order in which they are used in the results.
2. Fig 1 a bottom > this is final OD600nm, i.e. measured just before the transfer. This needs to be clear in the figure legend and caption.
3. Fig 1b: choose shorter and more explicit names for the clones characterized. It is important that the reader can identify and compare rapidly the ancestor, clones from day 7 and clones from day 10.
4. L123-126: the reference is to supplementary table 5 and not 4 (or more precisely, Table S4 and S5 should be in reversed order). From current table S5, it is clear that evolution in increasing concentrations of TCS has led to increased resistance to ciprofloxacin, cefotaxime and Fosfomycin but not to the other antibiotics tested and this should be also reported in the results.
5. Fig 2b: gene names seem to be written in to different polices + it would be useful to highlight genes involved in Ab resistance (those represented in fig 2c).
6. L178 should read "In addition to the changes in the expression of genes involved in ABR, ...". If I understand correctly the down regulation of defense mechanisms is also a result from the transcriptomic analysis and figure 3a is a sub-part of figure 2b with different genes annotated.
7. L199-200 should read "For donors, we used four E. coli MG1655 carrying each one of four different AMR plasmids"
8. L204: "and" should be removed.
9. L209-214: redundancies
10. L233: figS2 does not show this.
11. L244: figS4 and not S3 should be referenced here.
12. L233-244: again nsrR KO are not a good proxy for the effect of mutations obtained in experimental evolution.
13. L262: "least" should be changed to "the least" or "less".
14. L266-268: this sentence is grammatically incorrect
15. L342: remove one "be"
16. L424: verb missing
17. L446: "sequenced" should be "sequencing"
18. L874: "one replicate" should be "replicate 1".
19. L911-912: what does "flow plots two independent replicates were performed" mean?

Reviewer #2 (Remarks to the Author):

I am happy to inform you that after evaluation, the authors have satisfactorily addressed all of my concerns, resulting in a significant improvement in the quality of this almost-new submission. As a result, I no longer have any objections to the manuscript's publication.

Reviewer #3 (Remarks to the Author):

While the authors have conducted additional analysis to address reviewers' comments, they failed to address the most critical limitation raised for the original submission (also noted by other reviewers).

The potentially most interesting aspect of the work is the increased HGT in the evolved clones. However,

the quantification of this trait is not convincing due to the prolonged culturing (16hrs) before measuring transconjugants. The issue is not only whether this was done using plating or flow, but the very nature of prolonged culturing. The authors provided more measurements on additional clones but none of these addressed the critical technical limitation. On that note, the flow measurements are not clearly described with regard to how it's calibrated. The corresponding figure panels are missing labels. Therefore, I am not confident in this conclusion. And I think the significance of the work rests upon the demonstration of this point.

Additionally, they used a competition assay to measure the plasmid fitness, which is not suitable in this case as the plasmids are transferable. The relative abundances after the "competition" confound both growth competition and potential gene transfer during this time window.

There are also other issues with the manuscript both in clarity of presentation and rigor of analysis. The text remains to be difficult to read. The reporting of statistical analysis is also quite sloppy. For example, at least in one case, they reported an exceedingly small p value of 5.76×10^{-108} , which is a recognized poor practice.

REVIEWER COMMENTS

Reviewer #1 (Remarks to the Author):

I had previously reviewed the manuscript by Yang and collaborators when it had been submitted to [Redacted]. This new version represents a clear improvement and the authors have performed additional experiments, redone analyses and deeply rewritten the manuscript. However, the general impression of lack of rigour is still present, although to a lesser extent than before. I list below the points that need to be addressed.

Response: we thank reviewer#1 for very useful comments. In accordance with all reviewers' suggestions, we have thoroughly revised the manuscript to fully address all reviewers' comments. To add more rigor to our findings, we have made the required modifications in the main text, and extensively revised the key figures (Fig.1-Fig.4). We also provide more robust evidence for the increased plasmid permissiveness of triclosan resistant mutants, by measuring the plasmid transfer at different time points cross 24 hours using flow cytometry and confocal microscopy. In the revised manuscript, changes in the main text were all tracked and the detailed point-by-point rebuttal are listed below.

1. The novelty of the paper mainly resides in the increased conjugation rates and phage sensibility of the populations evolved in presence of triclosan and the exploration of a potential mechanistic explanation for this pleiotropic effect. The part on TCS resistance evolution and evolution of cross-resistance to other antibiotics could be shortened to get more rapidly to the most interesting and newest point. This finding should also be discussed in the context of other recent studies on similar questions (e.g. Xuan et al. 2022 doi: 10.1128/spectrum.01356-22; Chen et al. <https://www.nature.com/articles/s41598-017-06688-w>; Chan et al. <https://www.nature.com/articles/srep26717>).

Response: we have tried earnestly to reduce the text in accordance with reviewer's comments without compromising the rigor of data and analysis. as suggested, the results of TCS resistance evolution and mutations have been substantially shortened (L130-177). We have also re-arranged the figures (Fig.1-4) and showed the increased conjugation rates in Fig.2-3, and phage sensibility showed in separate figures (Fig.4 and supplementary Fig.S6-8)

The two suggested references have been discussed in the Discussion (L373-374).

2. Whole genome sequencing: there is still a lot of confusion about the number and the identity of the clones sequenced. The results section says "five clones from day 7" and then mentions "other clones from day 8 and 10. Figure 2 legend says "five from day 7 + two from day 10". Methods says "13 TRMs from day 7 to 10" which corresponds to the content of table S4. Even more importantly, it is still not clear to me whether clones from different

days are sampled from independent experimentally evolved populations. The interpretation of finding the same mutations in clones from different time points, although informative in both cases, is quite different depending on whether they come from the same population or not.

Response: we apologize for the confusion in how we describe the selection of evolved clones. Here, we would like to provide a gantt chart to clarify the number and the identity of the evolved clones sequenced. As described in method (L442) and the below table, in total we sequenced 13 evolved clones isolated across different time points (Table S4).

To avoid any overstatement of gene mutations in the evolved clones (in light of previous reviewers' comments), we chose five evolved clones from independent replicates at day7 and two final evolved clones from day10 for our conjugation and phage susceptibility experiments (Fig.2-4, L147-149).

We hope we have provided a clearer explanation for how the number and identity of the evolved clones were chosen.

Replicates	day 0	day 6	day 7	day 8	day 9	day 10	day 11
TCS(mg/l)	0	1 (2xMIC)	2 (4xMIC)	4 (8xMIC)	8 (16xMIC)	16 (32xMIC)	32 (32xMIC)
replicate-1	d0-1	d6-1	d7-1	d8-1	x	x	x
replicate-2	d0-2	d6-2	d7-2	d8-2	d9-2	d10-2	x
replicate-3	d0-3	d6-3	d7-3	d8-3	d9-3	d10-3	x
replicate-4	d0-4	d6-4	d7-4	d8-4	x	x	x
replicate-5	d0-5	d6-5	d7-5	x	x	x	x
replicate-6	d0-6	d6-6	d7-6	x	x	x	x
replicate-7	d0-7	d6-7	d7-7	x	x	x	x
replicate-8	d0-8	d6-8	d7-8	x	x	x	x
replicate-9	d0-9	d6-9	x	x	x	x	x
replicate-10	d0-10	d6-10	x	x	x	x	x
replicate-11	d0-11	d6-11	x	x	x	x	x
replicate-12	d0-12	x	x	x	x	x	x
replicate-13	d0-13	x	x	x	x	x	x
replicate-14	d0-14	x	x	x	x	x	x
replicate-15	d0-15	x	x	x	x	x	x
Survival No.	15	11	8	4	2	2	3

evolved strains highlighted in red indicate the strains were sequenced; "X" indicates dead

3. L225-226: where does this affirmation come from? Please refer to a figure, a table, a statistical test + “singularly responsible” should “responsible alone” + as mentioned in my first review, the mutations in nsrR in exp evol populations are non-synonymous SNPs so there was no reason to expect that KO would recapitulate the effect of these mutations.

Response: This sentence has been rewritten (L231-233).
“responsible alone” has been corrected (L239).

Although we failed to generate the point mutation on nsrR after numerous attempts, our results strongly suggest that nsrR can modulate the conjugation permissiveness to AMR plasmids and phage sensibility. As queried by reviewer#3, in this revised manuscript, we have repeated one conjugation experiment with pFII_mcr-8 plasmid, to measure the conjugation rates at different time points (revised Fig.3). The evolved d10-2 and nsrR

knockout strain showed significantly increased conjugation rates after 8 hours co-incubation (Fig.3 and supplementary Fig.S5).

Furthermore, in the accordance with reviewers' comments, we have addressed this issue as a limitation in the Discussion (L388-390).

4. L114-115 “re-exposed to 2 and 4mg/L of TCS in short-term growth assays.” + FigS1: I do not understand the interest of getting this growth curves as the clones have been isolated at days where the TCS concentration was equal or superior to 4mg/L, so it is clear that they are able to grow at these two concentrations. The only piece of info from figure S1 is that TCS resistance has a cost in absence of TCS (TRMs grow less than the ancestor at 0mg/L), but this is not mentioned neither in the results nor in the discussion. More specifically, on 1114-115, no results are actually given for this specific assay and as the following sentence is about MIC, it seems that the MIC have been obtained by this “short-term growth assay” but this is technically not possible (broader concentration range needed), so it is inducing confusion.

Response: this sentence has been rewritten (L113-116), the MIC of triclosan was determined by agar dilution protocol, which has also mentioned in the method section (L421-431). We have replaced the supplementary Fig.S1 with the growth curves of nsrR and ndh knockout strains.

Minor points

These points are minor but all together they contribute to the impression of lack of rigour.

1. The methods are not presented in the order in which they are used in the results.

Response: we have added methodological details for conjugation experiment measuring by flow cytometry and microscopy (L574-585), as well as re-arranged the methods section, to chronologically present the methodological approach.

2. Fig 1 a bottom > this is final OD_{600nm}, i.e. measured just before the transfer. This needs to be clear in the figure legend and caption.

Response: final OD_{600nm} has been revised throughout the manuscript. In the Methods, we have clarified that bacterial density was measured before each transfer (L426-428).

3. Fig 1b: choose shorter and more explicit names for the clones characterized. It is important that the reader can identify and compare rapidly the ancestor, clones from day 7 and clones from day 10.

Response: the names of evolved strains have been shortened by removal of strain ID (Kp85-). the naming code is based on time-points and replicates. For example, d10-2 name indicates that the evolved clone was isolated from day 10 in replicate 2).

4. L123-126: the reference is to supplementary table 5 and not 4 (or more precisely, Table S4 and S5 should be in reversed order). From current table S5, it is clear that evolution in increasing concentrations of TCS has led to increased resistance to ciprofloxacin,

cefotaxime and Fosfomycin but not to the other antibiotics tested and this should be also reported in the results.

Response: The order of Table S4 and S5 have been reversed. the clarification of no change to the other antibiotics tested was also added in L127-128.

5. Fig 2b: gene names seem to be written in to different polices + it would be useful to highlight genes involved in Ab resistance (those represented in fig 2c).

Response: As there were over 400 genes differentially expressed in evolved d10-2 strain, it is difficult to highlight the Ab resistance genes in Fig.S2c (moved to the supplementary Figure), this is also the reason for providing the Fig.S4 (moved to the supplementary Figure), specifically listing the Ab and TCS resistance genes.

6. L178 should read “In addition to the changes in the expression of genes involved in ABR, ...”. If I understand correctly the down regulation of defense mechanisms is also a result from the transcriptomic analysis and figure 3a is a sub-part of figure 2b with different genes annotated.

Response: this sentence has been corrected.

7. L199-200 should read “For donors, we used four E. coli MG1655 carrying each one of four different AMR plasmids”

Response: this sentence has been corrected.

8. L204: “and” should be removed.

Response: “and” has been deleted in L216.

9. L209-214: redundancies

Response: these sentences were deleted.

10. L233: figS2 does not show this.

Response: Fig.S1 was added here (L248)

11. L244: figS4 and not S3 should be referenced here.

Response: corrected

12. L233-244: again nsrR KO are not a good proxy for the effect of mutations obtained in experimental evolution.

Response: we have added the limitation in the discussion (L388-390).

13. L262: “least” should be changed to “the least” or “less”.

Response: corrected in L264

14. L266-268: this sentence is grammatically incorrect

Response: This sentence has been rewritten (L287-288)

15. L342: remove one “be”

Response: “be” was removed from this sentence L344.

16. L424: verb missing

Response: this sentence was corrected in L451.

17. L446: “sequenced” should be “sequencing”

Response: sequencing was corrected in L473

18. L874: “one replicate” should be “replicate 1”.

Response: corrected in Figure 2 legend, L919

19. L911-912: what does “flow plots two independent replicates were performed” mean?

Response: this sentence was correct in Fig.4 figure legend. L956

Reviewer #2 (Remarks to the Author):

I am happy to inform you that after evaluation, the authors have satisfactorily addressed all of my concerns, resulting in a significant improvement in the quality of this almost-new submission. As a result, I no longer have any objections to the manuscript's publication.

Response: Thank you for these very positive comments. we are delighted with the positive response and the reviewer’s appreciation of the significance and potential impact of this article.

Reviewer #3 (Remarks to the Author):

While the authors have conducted additional analysis to address reviewers' comments, they failed to address the most critical limitation raised for the original submission (also noted by other reviewers).

Response: we thank reviewer#3 for very useful comments. In accordance with all reviewers’ suggestions, we have thoroughly revised the manuscript to fully address all reviewers’ comments. To add more rigor to our findings, we have made the required modifications in the main text, and four main figures (Fig.1-Fig.4) have been extensively revised. We also provide more robust evidence for the increased plasmid permissiveness of triclosan resistant mutants, by measuring the plasmid transfer in different time points across 24 hours using flow cytometry and confocal microscopy. In the revised manuscript, changes in the main text were all tracked and the detailed point-by-point response are listed below.

The potentially most interesting aspect of the work is the increased HGT in the evolved clones. However, the quantification of this trait is not convincing due to the prolonged culturing (16hrs) before measuring transconjugants. The issue is not only whether this was done using plating or flow, but the very nature of prolonged culturing. The authors provided more measurements on additional clones but none of these addressed the critical technical limitation. On that note, the flow measurements are not clearly described with regard to how it's calibrated. The corresponding figure panels are missing labels. Therefore, I am not confident in this conclusion. And I think the significance of the work rests upon the demonstration of this point.

Response: To add more robust evidence to our findings, we have performed time-course conjugation experiment where microscopy images/flow plots of the conjugation population were acquired at different time points (0, 2, 4, 6, 8, 12, 16 and 24 hours, see the Method L574-585). It is noted that no plasmid transfer occurred until 4 hours after mixing donor-recipient cells, and the number of transconjugants (green cells) was significantly increased in evolved strain d10-2 and nsrR KO strain after 8-hour mating ($p < 0.05$), comparing to the parental Kp85anc strain (the new Fig.3 and supplementary Fig.S5).

The method of measuring plasmid transfer by flow cytometry was inspired by previous studies that determining the RP4 plasmid transfer in bacterial populations (ref. Soren J. Sorensen et al. 2005 Nature Reviews Microbiology, Studying plasmid horizontal transfer in situ: a critical review | Nature Reviews Microbiology). In the new Fig.3a, flow plots were provided and it shows clearly that conjugation permissiveness toward pFII-mcr-8 plasmid was significantly increased in both nsrR knockout and d10-2 strain. The percentage of gfp-expressing transconjugants was also highlighted in red circle in each plot.

Fig.3 Increased conjugation rates of AMR plasmids were constantly observed in evolved and $\Delta nsrR$ knockout clones across 24 hours. (measured by flow cytometry, the full figure legend was provided in the main text)

Fig.S5 Increased conjugation rates of AMR plasmids were constantly observed in evolved and $\Delta nsrR$ knockout clones across 24 hours. (measured by confocal laser microscopy, the full figure legend was provided in the supplementary figure S5)

Additionally, they used a competition assay to measure the plasmid fitness, which is not suitable in this case as the plasmids are transferable. The relative abundances after the "competition" confound both growth competition and potential gene transfer during this time window.

Response: Whilst this competition model is well defined in microbiology, we appreciate the limitation of these assays and agree with the reviewer that this method can't prevent the transfer of plasmid during the competition assay. This additional data does not affect our main findings in increasing HGT for plasmids and phage susceptibility, therefore, we have removed this analysis from the Methods and supplementary data.

There are also other issues with the manuscript both in clarity of presentation and rigor of analysis. The text remains to be difficult to read. The reporting of statistical analysis is also quite sloppy. For example, at least in one case, they reported an exceedingly small p value of 5.76×10^{-108} , which is a recognized poor practice.

Response: We have tried earnestly to revise the main text and figures in accordance with reviewers' comments. Four main figures have been extensively revised, especially adding more data in Figure 3 and 4.

The small p values were generated from the RNAseq data analysis, and differential gene expression analysis was conducted by the edgeR package (v3.40.2)(ref.64, Robinson et al., 2010, as showed in the Reference). This method has been well acknowledged for examining differential expression of replicated count data. The gene expression changes between the evolved clone and the parental clone were very significant, resulting in a small p -value.

The other data analysis was performed using GraphPad Prism (8.3.1). Data shown in plots are represented as mean of at least two replicates \pm SEM, and exact number of independent replicates for each experiment is stated in their respective figure legends. Holm-Sidak or Wilcoxon t-test analysis ($p < 0.05$) was used to compare differences on conjugation experiments (see in the Methods L692-697).

Reviewers' comments:

Reviewer #3 (Remarks to the Author):

I don't think the presented time courses address the central criticism I had on the paper. That is, I am not convinced by the data presented that conjugation rates have increased in the evolved strains or the knocked-out strains. They may have increased but the data are inconclusive.

Specifically, as pointed out previously, the critical issue is not whether time course data are presented. Rather, it's that the experimental protocol for measuring conjugation rates is fundamentally flawed. Over the course of 24 hours (or 6-8hrs when they saw some effects), even slight differences in the growth rates of different strains can drastically affect the relative fractions of transconjugants, even if there is no change in the conjugation rate.

The nuances of various techniques to measure conjugation rates have been well recognized and discussed in detail in Huisman et al, *Plasmid* 2022. When growth is entwined with conjugation, it is especially difficult to provide a good estimate of conjugation rates or even to draw conclusions on whether conjugation rates have increased. While the authors conducted extensive measurements, these measurements are not well designed to determine if conjugation rates have or have not increased in various strains.

On a more minor point, I understand how the p-values are generated from the use of statistic software. However, reporting exceedingly small p-values gives a false sense of certainty and potentially masks more important control parameters, an issue that was recently discussed in Huber, *Cell Syst.* 2019.

Reviewer #4 (Remarks to the Author):

I was asked to review this paper as an arbitrating reviewer to comment on the outstanding concerns of Reviewer #1 and Reviewer #3.

Regarding Reviewer #3 concerns, I agree that it would be better to present conjugation rates instead of conjugation frequencies. Conjugation rates correct for the potential biases introduced by the difference in growth rates of donors, recipients and transconjugants. There are different methods to calculate conjugation rates, such as the end point method described by Simonsen et al. (DOI: 10.1099/00221287-136-11-2319) or other more modern methods described in Huisman et al. (DOI: 10.1016/j.plasmid.2022.102627).

Moreover, I'm afraid I also tend to agree with Reviewer #1 about his/her concerns leading to a "general impression of lack of rigour".

Point-to-point answers to all comments:

Reviewers' comments:

Reviewer #3 (Remarks to the Author):

I don't think the presented time courses address the central criticism I had on the paper. That is, I am not convinced by the data presented that conjugation rates have increased in the evolved strains or the knocked-out strains. They may have increased but the data are inconclusive.

Specifically, as pointed out previously, the critical issue is not whether time course data are presented. Rather, it's that the experimental protocol for measuring conjugation rates is fundamentally flawed. Over the course of 24 hours (or 6-8hrs when they saw some effects), even slight differences in the growth rates of different strains can drastically affect the relative fractions of transconjugants, even if there is no change in the conjugation rate.

The nuances of various techniques to measure conjugation rates have been well recognized and discussed in detail in Huisman et al, *Plasmid* 2022. When growth is entwined with conjugation, it is especially difficult to provide a good estimate of conjugation rates or even to draw conclusions on whether conjugation rates have increased. While the authors conducted extensive measurements, these measurements are not well designed to determine if conjugation rates have or have not increased in various strains.

Response: Firstly, we would like to apologise that we did not fully understand reviewer's request on conjugation rate experiment in the previous round of revision i.e. that we need to estimate the conjugation rate using the Simonsen's end-point method. We have now conducted additional experimental work to fully and unequivocally address these points.

To address reviewer's comments, we spent several weeks repeating all the conjugation rate experiments using the suggested Simonsen's endpoint calculation method (added in the methods, L516-528). In this "end-point" method, several key parameters have been included to estimate the conjugation rates, e.g. conjugation time and the initial/final population density of measurement. Our new results are almost identical with the conjugation estimation method we originally described in our article, providing *prima facie* evidence for the involvement of triclosan resistance mutants (TRMs) in broad AMR plasmid permissiveness (new data Fig.2c-d and Fig.3b, showed in the above rebuttal text). Moreover, we have performed new growth curves for the evolved strains and plasmids and found no significant difference between these and ancestral Kp85anc strain (new data Fig.S1), even though evolved strains had slightly reduced carrying capacity. As a result, changes in conjugation rate were an unlikely explanation for the differences observed in growth rates between the evolved TRMs and ancestral Kp85anc strains.

Fig.2c-d (new data, revised in the main Fig.2c-d) showing the transfer of four AMR plasmid in seven evolved strains and Kp85anc strain, based on Simonsen's end-point method. The different plasmid types were distinct with different colors, and the bacterial density were also measured by two methods, flow cytometry and selective agar plating.

The previous figure Fig.2c-d for the comparison with the above new data. This data was presented in the main text of previous version (Fig.2c-d). Here, the transfer rates were calculated using the ratio of transconjugant cells in total bacterial cells or recipient cells.

The results are highly similar with the new method presented above (new data Fig.2c-d).

Fig.S1 (new data) comparing the growth rates between evolved strains and ancestral Kp85anc strain.

Furthermore, we have also repeated the experiment and calculated the time-course transfer dynamic of pFII_MCR-8 plasmid using Simonsen's end-point method (**new data Fig. 3b below**). These new results support the key conclusion that the evolved strain and nsrR knockout showed increased conjugation permissiveness towards plasmid pFII_MCR-8.

Simonsen' end-point calculation

Previous calculation method (the ratio between tranconjugant and donor cells)

(the top row of the figure, new data Fig.3b) showing the transfer rate of pFII-MCR-8 plasmid calculated by Simonsen's end-point method.

(the bottom figure; the previous figure Fig.3b) for the comparison with the above new data, showing the transfer rate of pFII-MCR-8 plasmid calculated by the classic method

(ratio between the number of tranconjugants and recipient cells).

On a more minor point, I understand how the p-values are generated from the use of statistic software. However, reporting exceedingly small p-values gives a false sense of certainty and potentially masks more important control parameters, an issue that was recently discussed in Huber, Cell Syst. 2019.

Response: Whilst the edgeR method has been recognized as a powerful and efficient tool for differential expression analysis of RNA-seq data (Baltoni, Nucleic Acid Res. 2023; Robinson, Bioinformatics. 2010), we appreciate the limitation of this assay which may give a very small P values. However, in our main text, we have designed experiments to verify the results observed in RNAseq data, for instance, the MIC test to confirm the increased/decreased expression of several resistance determinants (Fig.1e, L181-187). And the down-regulated bacterial defense system can help to explain our main findings in increasing HGT for plasmids and phage susceptibility.

Other statistical analysis (main Fig.1-4) were performed by prism software and all P values were indicated in the figures. In addition, we have now rounded up all the small P -values and report them as $P < 0.001$ throughout the text.

Reviewer #4 (Remarks to the Author):

I was asked to review this paper as an arbitrating reviewer to comment on the outstanding concerns of Reviewer #1 and Reviewer #3.

Regarding Reviewer #3 concerns, I agree that it would be better to present conjugation rates instead of conjugation frequencies. Conjugation rates correct for the potential biases introduced by the difference in growth rates of donors, recipients and transconjugants. There are different methods to calculate conjugation rates, such as the end point method described by Simonsen et al. (DOI: 10.1099/00221287-136-11-2319) or other more modern methods described in Huisman et al. (DOI: 10.1016/j.plasmid.2022.102627).

Moreover, I'm afraid I also tend to agree with Reviewer #1 about his/her concerns leading to a "general impression of lack of rigour".

Response: To address reviewer's comments, we spent several weeks repeating all the conjugation rate experiments using the suggested Simonsen's endpoint calculation method. In this "end-point" method (added in the methods, L516-528), several key parameters have been included to estimate the conjugation rates, e.g. conjugation time and the initial/final population density of measurement. Our new results are almost identical with the conjugation estimation method we originally described in our article, providing *prima facie* evidence for the involvement of triclosan resistance mutants (TRMs) in broad AMR plasmid permissiveness (new data Fig.2c-d and Fig.3b, showed in the above figures). Moreover, we have performed new growth curves for the evolved strains and plasmids and

found no significant difference between these and ancestral Kp85anc strain (Fig.S1, showed in the above rebuttal text), even though evolved strains had slightly reduced carrying capacity. As a result, changes in conjugation rate were an unlikely explanation for the differences observed in growth rates between the evolved TRMs and ancestral Kp85anc strains.

Reviewers' comments:

Reviewer #3 (Remarks to the Author):

In the revised analysis, the authors used Simonsen's endpoint method to compute the transfer rate. This is one of the several methods reviewed in the Huisman paper. This analysis is in the right direction by considering the growth of the three populations.

However, while I suggested the Huisman paper as summary of various issues regarding the computation of transfer rates, I did not suggest the use of Simonsen's method. For reasons explained below, Simonsen's method may not be the optimal method and the authors were using it improperly.

Simonsen's method assumes the same growth rate for all three populations (donor, recipient, and transconjugant).

It also assumes the endpoint is still during the exponential phase.

In the methods section in the revision, I don't see how the growth rates were computed and how these assumptions were justified. From Fig S1, the different strains still have somewhat different growth rates (which should be computed during the exponential phase). From these curves, it is clear that the exponential phase has ended before 4h.

The authors computed the transfer rates using multiple endpoints (4h, 8h, etc). The ones using later endpoints directly violated the assumptions of Simonsen's method and should be removed, even if they can somehow justify the first assumption.

Point-to-point response to all comments:

Over several rebuttals we have made various iterations of our manuscript and on every occasion have responded fully to the reviewer's suggestions - both in the paper and our point-point response. The outstanding debate is whether we have fully addressed the issue of conjugations rates? Below are the final comments from reviewer #3 which we have continued to fully address.

Reviewer #3 (Remarks to the Author):

In the revised analysis, the authors used Simonsen's endpoint method to compute the transfer rate. This is one of the several methods reviewed in the Huisman paper. This analysis is in the right direction by considering the growth of the three populations.

However, while I suggested the Huisman paper as summary of various issues regarding the computation of transfer rates, I did not suggest the use of Simonsen's method.

Response: In our previous revision, we sort to address the issues mainly raised by reviewer #3 and reviewer #4 (and, apparently, by deferment reviewer #1 although they did not specify their concerns). We were specifically given two references to refer to - Simonsen et al. (DOI: 10.1099/00221287-136-11-2319) and Huisman et al. (DOI: 10.1016/j.plasmid.2022.102627) – the latter, describes three variations on the original Simonsen model (page 4; 3.1 “Simonsen model (SM)”; page 4; 3.2 “Extended Simonsen model (ESM)” and page 5; 3.3 “The approximate Extended Simonsen model (ASM)”). Ergo, they are all Simonsen models using the same parameters (conjugation time, bacterial growth rate, initial/final bacterial density, etc.) and contrary to the reply of Reviewer #3, we were explicitly instructed by the reviewers to apply the Simonsen model which is exactly what we have done.

Reviewer#4 has specifically suggested to use Simonsen's end-point method (*There are different methods to calculate conjugation rates, such as the end point method described by Simonsen et al. (DOI: 10.1099/00221287-136-11-2319) or other more modern methods described in Huisman et al. (DOI: 10.1016/j.plasmid.2022.102627)*); accordingly, we carried out many additional conjugation experiments and conjugation rates were calculated by applying Simonsen's across different time points. Our results show no appreciable differences from our previous analysis and certainly do not impact on the main findings of our paper.

For reasons explained below, Simonsen's method may not be the optimal method and the authors were using it improperly. Simonsen's method assumes the same growth rate for all three populations (donor, recipient, and transconjugant).

It also assumes the endpoint is still during the exponential phase.

Response: Firstly, according to the original paper (Simonsen et al.,1990; doi: 10.1099/00221287-136-11-2319) (final paragraph of page 2), the Simonsen method (SM) can be applied at any time point during the experiment and the maximum growth rate (ψ_{\max}) was calculated from conjugation populations during the phase of exponential population growth. According to Huisman et al., mating cultures were also incubated for 24h. We have repeated many conjugation assays that fully align with the descriptions given in the two references (now

described in our revised paper by lines 524-561), where ψ_{max} was obtained at 4h. The time course growth rate of parental and evolved strains was

$$transfer\ rate\ (\gamma) = \psi_{max} \cdot \ln\left(1 + \frac{T \cdot N}{R \cdot D}\right) \cdot \left(\frac{1}{N - N_0}\right)$$

Secondly, we also applied Simonsen's derived method (the approximate Extended Simonsen model, ASM), which caters for different bacterial growth rates by D, R and T strains. We used the provided online calculator to estimate the conjugation rates (Huisman et al. page 10, "5. Tools for the scientific community": <https://ibz-shiny.ethz.ch/jhuisman/conjugator/>) (we have added this in the main text of our article in lines 528-529). The conjugation rates calculated by ASM are almost identical with the Simonsen end-point method (SM), which consistently show that triclosan resistance mutants (TRMs) have enhanced AMR-plasmid permissiveness (showed in the below - figure 1). As both calculations show almost identical results, we selected the classic Simonsen's end-point method (SM) in our revised manuscript which are illustrated in Fig.2 and Fig.3. It is important to note that Huisman et al. compared original Simonsen and the extended Simonsen methods, and both conjugation rates were shown to be nearly identical at 4 and 8h (Fig.3, SM end point and ASM end point).

Most importantly, our manuscript has now undergone four rounds of comments and rebuttals, and we have been earnest in applying all available methods and assays to determine the conjugation rates (lines 477-555, also see below Table 1), including classic T/R calculation, Simonsen's end-point method (SM) and ASM calculations. Bacterial density was also measured by selective agar plating, fluorescence-activated flow cytometry, confocal microscopy and microplate readers. The time-course conjugation experiments also have been monitored by flow cytometry and confocal microscopy (lines 507-520). The exact conjugation rates estimated by different methods were also provided in the supplementary Excel files (added in the main text lines 205-207; 228-229; and supplementary Fig.S5-S8).

Table 1 measures of conjugation rates performed in this study. the calculation formula for each conjugation assay has been added in the main text (lines 477-550)

Mating culture	Measure of bacterial density	Measure of conjugation rates
Liquid	Selective agar plating (CFU/ml)	Conjugation rates per recipient (T/R)
Liquid	Flow cytometry (CFU/ml)	Conjugation rates per cell (T/N) (plasmids were tagged with gfp gene ¹)
Liquid	Confocal microscopy (cell counts)	Conjugation rates per donor (T/D) (plasmids were tagged with gfp gene ¹)
Liquid	Plating (CFU/ml) and microplate reader (OD ₆₀₀)	using SM, ASM calculators (Huisman et al.)

D, R, T stands for the bacterial density of donors, recipients, and transconjugants at the time point of measurement, N is the total population density ($N = D + R + T$), N_0 is the initial total population density, ψ_{\max} is the maximum growth rate of the mating culture.

All of these datasets consistently demonstrated that the TRMs have improved conjugation permissiveness (see below supplementary Figure S5-S8, linked to main Fig.2d, Fig.2e, Fig.3b and Fig.3c-d, respectively). We also provide all conjugation results in supplementary excel file. However, these data, apparently, still appear not to meet this reviewer's requirements despite the fact we have followed, to the letter, his/her advice and the advice of reviewer #4.

supplementary Fig.S5 (linked to main Fig.2d) The comparison of conjugation rates measured by three different methods. The exact conjugation rates were also provided in the supplementary Excel files.

supplementary Fig.S6 (linked to main Fig.2e) The comparison of conjugation rates measured by three different methods. The exact conjugation rates were also provided in the supplementary Excel files.

supplementary Fig.S7 (linked to main Fig.3b) The comparison of conjugation rates measured by three different methods. The exact conjugation rates were also provided in the supplementary Excel files.

supplementary Fig.S8 (linked to main Fig.3c-d) The comparison of conjugation rates measured by three different methods. The exact conjugation rates were also provided in the supplementary Excel files.

Whilst, predictably, there are variations in the datasets using different assays and calculation, the original premise of our article, the effects and impact of evolved strains, remains the same.

In the methods section in the revision, I don't see how the growth rates were computed and how these assumptions were justified. From Fig S1, the different strains still have somewhat different growth rates (which should be computed during the exponential phase). From these curves, it is clear that the exponential phase has ended before 4h.

Response: Firstly, the growth profiles of each individual strain, D, R and T, and mixed conjugation culture were estimated by growth assays using 96-well microtiter plate and bacterial density were measured as the optical density at 600 nm (OD_{600nm}) every hour by SpectraMax iD3 (Molecular Devices, United States) (described in our method section- Lines 363-365). The bacterial growth rate ψ was determined by the optical density of each culture at different time points, where T_b and T_a indicate the time s (h) at which OD_b and OD_a (described in our revised manuscript - Lines 551-561). ψ_{max} was used to calculate the transfer rates.

The authors computed the transfer rates using multiple endpoints (4h, 8h, etc). The ones using later end-points directly violated the assumptions of Simonsen's method and should be removed, even if they can somehow justify the first assumption.

Response: We feel that the word "violate" is deliberately inflammatory, inappropriate and incorrect. According to Huisman et al. (DOI: 10.1016/j.plasmid.2022.102627), this method can be applied at any time point during the experiment (Page 4, "3.1 The Simonsen method") which

is supported in the original paper (Simonsen et al.,1990; doi: 10.1099/00221287-136-11-2319) (final paragraph of page 2): “The cultures may be sampled at any time during exponential growth or stationary phase (provided there is no cell death or plasmid transfer after growth ceases), when the transconjugant population density is large *Estimating plasmid transfer rates enough to be detected.*”

Reference for previous published work by this group using flow cytometry and confocal microscopy.

1. Yang, Q. E. *et al.* Interphylum dissemination of NDM-5-positive plasmids in hospital wastewater from Fuzhou, China: a single-centre, culture-independent, plasmid transmission study. *The Lancet Microbe*, doi:10.1016/s2666-5247(23)00227-6 (2023).

REVIEWER COMMENTS

Reviewer #5 (Remarks to the Author):

I was asked here to comment specifically on concerns about the methodology relating to conjugation rates, and remaining concerns of reviewers 3 and 4.

On the specific point of which formulas to use to calculate conjugation, I do agree with the authors. They have followed reviewers' requests to use Simonsen's endpoint model or its corrections. Prolonged culturing is not a problem as even the original Simonsen's endpoint model does not assume that the endpoint is still during the exponential phase; differences in growth rate are accounted for by the extended Simonsen's model described in Huisman et al and used here.

However, I do agree with the reviewers' worry about the data being inconclusive/unclear.

First I suspect this is partly due to the manuscript having gone through many revisions and reshuffling, it would need to be read again carefully with fresh eyes for global coherence (as an example, Fig 2 title only fits 2a and maybe b but not c to e). This also affects the conjugation rates results and methods: the method section does mix experimental methods (flow cytometry, plating, microscopy) and formulas for conjugation rate in a very confusing way. From the methods, plating data seem to still be analysed with T/N, flow cytometry by T/N and microscopy by T/D. The results shown in the main manuscript are sometimes summary data, sometimes detailed cytometry plots that I believe should all go in the supplementary data.

The results also vary depending on which experimental method is used, and this is not discussed. I personally would not have asked for this many different methods, but as they give very different results, this needs at least mentioned. Figs 2d and 2e differ by around 3 orders of magnitude for the same plasmid: What could explain this, if all methods reliably evaluate donors, recipients and transconjugants? The unit of measurement is also lacking, making it hard to evaluate these rates (or to guess which calculation has been used).

There are also a few worrying facts in the plots: Fig 3c pX3-NDM y scale has 3. 10⁻¹⁶ repeated twice; and some bars have 1 single replicate visible and yet an associated p-value (Fig S6).

A very large amount of data is now presented, but not all of them are discussed, instead transfer rates are claimed to be constantly / consistently higher, which is not always true. E.g. line 222, MCR-8 is claimed to have "constantly higher transfer rates in the evolved d10-2 clone" – this is not true at 12h as shown in Fig 3b. And at 16h from Fig 3a there are no more transconjugants either – maybe the calculated transfer rate is higher but then providing the raw data might help understand why – does the effect arise from less donors or recipients?

So any effect is not as consistent as the authors imply.

P-values are shown for each time point / strain, but with so many conditions tested it is not clear if the global effect (evolved clones vs ancestor, or at least d10-2 clone across timepoints) is significant, and with which method.

Finally, the raw data need to be available to the reader. What is presented now is only end estimates of the conjugation rate, but not the raw experimental data: donor, recipient and transconjugant densities (or proportions in the case of flow cytometry) need at least to be provided, plus the growth rates needed for Simonsen's method.

Overall, I believe the conjugation data are worth publishing - and do certainly not need more experiments with more methods! - but the existing data need analysed in a more streamlined way to make the possible patterns clearer.

Two added notes, even if it is further from what I was asked to comment on: i) the refs cited I254 seem to be about transformation, not conjugation. And ii) I believe that the general argument of this manuscript about evolution of permissiveness to MGEs via down regulation of defence systems could be also supported by insisting more on the phage data. Plaque assays are more straightforward to analyse in this context, as there are no donors to worry about and 'recipients' are in excess.

REVIEWER COMMENTS

Reviewer #5 (Remarks to the Author):

I was asked here to comment specifically on concerns about the methodology relating to conjugation rates, and remaining concerns of reviewers 3 and 4.

Response: Thank you very much for your time and for your oversight regarding the lengthy discussions involving our conjugation data and modelling conjugation rates/frequencies. Your valuable insights and comments have further improved the rigor of our findings, and we have addressed these concerns in our revised manuscript accordingly. We present below our point-by-point response:

On the specific point of which formulas to use to calculate conjugation, I do agree with the authors. They have followed reviewers' requests to use Simonsen's endpoint model or its corrections. Prolonged culturing is not a problem as even the original Simonsen's endpoint model does not assume that the endpoint is still during the exponential phase; differences in growth rate are accounted for by the extended Simonsen's model described in Huisman et al and used here.

Response: Thank you for confirming that the conjugation methods we used were appropriate and that we have diligently followed the reviewer's advice throughout these iterations. Since the conjugation frequencies have now been measured by at least five different calculation methods, we have described the results from original SM method in the main text and described the other calculations in the supplementary data (Fig.S5-S9). All the necessary bacterial parameters for the calculation of conjugation frequency have been compiled into the supplementary datasets 1-4. These datasets contain all raw experimental data (donor, recipient, and transconjugant densities), as well as growth rates required for the Simonsen's method.

However, I do agree with the reviewers' worry about the data being inconclusive/unclear.

Firstly, I suspect this is partly due to the manuscript having gone through many revisions and reshuffling, it would need to be read again carefully with fresh eyes for global coherence (as an example, Fig 2 title only fits 2a and maybe b but not c to e). This also affects the conjugation rates results and methods: the method section does mix experimental methods (flow cytometry, plating, microscopy) and formulas for conjugation rate in a very confusing way. From the methods, plating data seem to still be analysed with T/N, flow cytometry by T/N and microscopy by T/D. The results shown in the main manuscript are sometimes summary data, sometimes detailed cytometry plots that I believe should all go in the supplementary data.

Response: The manuscript has gone through many iterations in direct response to the

reviewer's suggestions; however, we unreservedly apologise if this has resulted in any confusion. To try providing further clarity on the key messages from our data, we have thoroughly revised both results (L184-222, L238-261) and method sections (L486-603). Further, we have undertaken the following:

(i) Firstly, the title of Fig.2 has been revised (L884): “*The genetic mutations and downregulation of bacterial defense systems leading to enhanced conjugation rate of AMR plasmids.*” We have also revised the figure legends for Fig.2 and Fig.3 as requested.

(ii) Secondly, since the conjugation frequency have been repeated numerous times with at last five different calculation methods (T/N, T/R, T/D, SM and ASM), we should clearly state the differences in each conjugation method. The choice of flow cytometry, plating or microscopy offered three different ways for quantifying the bacterial densities of the donor, recipient and transconjugants. We subsequently calculated the transfer frequency by using different bacterial growth parameters (e.g., time, bacterial growth rate, initial/final population density) obtained by the above methods. Therefore, to improve clarity and scientific rigor, we have re-organised the method section in a more streamlined manner, (L489-549), and the method for conducting time-course conjugation experiment (linked to Fig.3a-b) has been described in a separate paragraph (L583-L603). All related raw data and calculation file are available in supplemental datasets 1-4.

(iii) Thirdly, we have kept the results of transfer rates obtained from the original Simonsen's endpoint model (SM, the unit of transfer rate is $\text{mL} \cdot \text{cell}^{-1} \cdot \text{h}^{-1}$) in the main figures (Fig.2d-c and Fig.3b-c), while the conjugation frequency calculated by the other methods (T/N, T/R, T/D or ASM) we have described in the revised supplementary data - Fig.S5-S8. This is in accordance with the comments from reviewers' 3 and 4. Moreover, the main conjugation results have been revised thoroughly (L184-222, L238-261), and the detailed cytometry plots have been removed from Fig.3.

The results also vary depending on which experimental method is used, and this is not discussed. I personally would not have asked for this many different methods, but as they give very different results, this needs at least mentioned. Figs 2d and 2e differ by around 3 orders of magnitude for the same plasmid: What could explain this, if all methods reliably evaluate donors, recipients and transconjugants? The unit of measurement is also lacking, making it hard to evaluate these rates (or to guess which calculation has been used).

Response: Thank you for these observations and it is indeed interesting that different methods and calculations can produce variable data, and thus potentially affect any conclusions. It is anticipated that variations in bacterial density may arise when measured by two distinct methods. We have uploaded the raw data of all bacterial parameters (bacterial density, growth rates, OD values, etc.). The unit of measurement

was added in the y axis ($\text{mL} \cdot \text{cell}^{-1} \cdot \text{h}^{-1}$).

The main conjugation results have been revised thoroughly (L184-222, L238-265) to accommodate these differences. We appreciated the variations observed in plasmid transfer observed in different recipient-donor combinations, but it is also clear (Fig.2d-e and supplementary Fig.S5-S6) that the majority of the seven TRMs strains, have showed higher conjugation permissiveness compared to the parental strain and therefore, our overall conclusions, caveats notwithstanding, have not deviated from our original submission.

There are also a few worrying facts in the plots: Fig 3c pX3-NDM y scale has $3 \cdot 10^{-16}$ repeated twice; and some bars have 1 single replicate visible and yet an associated p-value (Fig S6).

Response: We apologise for this oversight. The reason why 3×10^{-16} appeared twice on the y axis is because we set the decimals as integers. We have now corrected this error in the revised Fig.3c. When we applied the ASM calculation from Huisman's suggested online conjugator ([doi:10.1016/j.plasmid.2022.102627](https://doi.org/10.1016/j.plasmid.2022.102627) 2022), some replicates did not give a value (indicated as NA), and that is the reason for the missing replicates in Fig.S6 ASM - pA/C-MCR-8.

A very large amount of data is now presented, but not all of them are discussed, instead transfer rates are claimed to be constantly/consistently higher, which is not always true. E.g. line 222, MCR-8 is claimed to have “constantly higher transfer rates in the evolved d10-2 clone” – this is not true at 12h as shown in Fig 3b. And at 16h from Fig 3a there are no more transconjugants either – maybe the calculated transfer rate is higher but then providing the raw data might help understand why – does the effect arise from less donors or recipients? So any effect is not as consistent as the authors imply.

Response: Thank you for your valuable feedback. We have revised our results analysis to provide a more measured description of the transfer rates, taking into account specific time points and variations observed from the experimental data. The main conjugation results have also been thoroughly revised (L184-222, L238-261). Moreover, we also compared the differences across timepoints using two-way ANOVA(L247-249), and confirmed that the transfer rates of pFII_MCR-8 were significantly affected by both mating times and *nsrR* variant strains. And the increased transfer rates were observed in at least four timepoints in both d10-2 and $\Delta nsrR$ strains (L249-255).

Additionally, we have provided raw data to accommodate a better understanding of any observed effects and their underlying causes, such as variations in donor or recipient abundance (supplementary datasets 1-4).

P-values are shown for each time point / strain, but with so many conditions tested it is not clear if the global effect (evolved clones vs ancestor, or at least d10-2 clone across timepoints) is significant, and with which method.

Response: The statistical analysis was performed by Holm-Sidak corrected two sampled *t*-test and *P*-values denote the statistical difference between each evolved strain and the ancestor kp85anc. The *p*-values were displayed on the corresponding bars (Fig.2 and Fig.3). The significant variability of plasmid transfer rate among TRM strains and ancestor was analyzed by one-way ANOVA (L203-208, and P values was added in a new supplementary Table S6), with the majority of TRM strains showing higher conjugation rates compared to the parental strain. Moreover, we also compared the differences across timepoints using two-way ANOVA(L247-249), and confirmed that the transfer rates of pFII_MCR-8 were significantly affected by both mating times and *nsrR* variant strains. And the increased transfer rates were observed in at least four timepoints in both d10-2 and $\Delta nsrR$ strains (L249-255).

Finally, the raw data need to be available to the reader. What is presented now is only end estimates of the conjugation rate, but not the raw experimental data: donor, recipient and transconjugant densities (or proportions in the case of flow cytometry) need at least to be provided, plus the growth rates needed for Simonsen's method.

Response: Thank you for your feedback regarding the availability of raw data. All the necessary bacterial parameters for the calculation of conjugation frequencies are now compiled into supplementary datasets 1-4. These datasets contain all the raw experimental data, including donor, recipient, and transconjugant densities, as well as growth rates required for Simonsen's method.

Overall, I believe the conjugation data are worth publishing - and do certainly not need more experiments with more methods! - but the existing data need analysed in a more streamlined way to make the possible patterns clearer.

Response: Thank you for your assessment of our manuscript and confirming that no additional experiments are needed. We fully acknowledge your suggestion to analyse the existing data in a more streamlined and coherent manner, and highlight key differences more clearly. We have re-analysed our conjugation data and have attempted to described it in a clearer and more concise way in both the results (L184-223, L238-265) and method sections (L499-605).

Two added notes, even if it is further from what I was asked to comment on: i) the refs cited L261 seem to be about transformation, not conjugation. And ii) I believe that the general argument of this manuscript about evolution of permissiveness to MGEs via down regulation of defense systems could be also supported by insisting more on the

phage data. Plaque assays are more straightforward to analyse in this context, as there are no donors to worry about and 'recipients' are in excess.

Response: Thank you for your additional notes. As rightly observed, these two references describe natural plasmid transformation; however, the basic parameters of plasmid transfer can still apply. Additionally, we acknowledge your suggestion regarding the phage data and the potential to further support the manuscript's message regarding the evolution of mobile genetic element permissiveness. We have utilised several plaque assays to strengthen our findings (Fig.4 and Fig.S10-S12) and also further emphasized the phage findings in the discussion (L364-372, 391-395).

REVIEWERS' COMMENTS

Reviewer #5 (Remarks to the Author):

The authors have satisfactorily addressed my comments, in particular the presentation of the conjugation assays is now much more straightforward to follow.

I only have a few reservations left about the abstract: i) TCS does not “increase the ability of *K. pneumoniae* to evolve resistance” (this would be changes in evolvability whereas what is shown here is simply selection of resistance), and ii) the sentence L43-47 seems to have acquired errors which were not in the previous version (increase instead of increased, ‘and’ has disappeared).

REVIEWERS' COMMENTS

Reviewer #5 (Remarks to the Author):

The authors have satisfactorily addressed my comments, in particular the presentation of the conjugation assays is now much more straightforward to follow.

I only have a few reservations left about the abstract: i) TCS does not “increase the ability of *K. pneumoniae* to evolve resistance” (this would be changes in evolvability whereas what is shown here is simply selection of resistance), and ii) the sentence L43-47 seems to have acquired errors which were not in the previous version (increase instead of increased, ‘and’ has disappeared).

Response: thank you very much for approving on our work. We greatly appreciate your time and comment. Your valuable insights and comments have further improved the rigor of our findings.

In our final version, we have revised the typo in the abstract by replacing “ability” to “evolvability” and adding “and” in L45.